# Empirical Study of Zero-Shot NER with ChatGPT

**Tingyu Xie**[1,2]**, Qi Li**[1,2]**, Jian Zhang**[1,2]**, Yan Zhang**[3*]**, Zuozhu Liu**[2]**, Hongwei Wang**[1,2*]

[1]College of Computer Science and Technology, Zhejiang University, China
[2]ZJU-UIUC Institute, Zhejiang University, China
[3]National University of Singapore, Singapore
{tingyuxie, liqi177, 12221038}@zju.edu.cn, eleyanz@nus.edu.sg
zuozhuliu@intl.zju.edu.cn, hongweiwang@zju.edu.cn

## Abstract

Large language models (LLMs) exhibited powerful capability in various natural language processing tasks. This work focuses on exploring LLM performance on zero-shot information extraction, with a focus on the ChatGPT and named entity recognition (NER) task. Inspired by the remarkable reasoning capability of LLM on symbolic and arithmetic reasoning, we adapt the prevalent reasoning methods to NER and propose reasoning strategies tailored for NER. First, we explore a decomposed question-answering paradigm by breaking down the NER task into simpler subproblems by labels. Second, we propose syntactic augmentation to stimulate the model's intermediate thinking in two ways: syntactic prompting, which encourages the model to analyze the syntactic structure itself, and tool augmentation, which provides the model with the syntactic information generated by a parsing tool. Besides, we adapt self-consistency to NER by proposing a two-stage majority voting strategy, which first votes for the most consistent mentions, then the most consistent types. The proposed methods achieve remarkable improvements for zero-shot NER across seven benchmarks, including Chinese and English datasets, and on both domain-specific and general-domain scenarios. In addition, we present a comprehensive analysis of the error types with suggestions for optimization directions. We also verify the effectiveness of the proposed methods on the few-shot setting and other LLMs.[1]

## 1 Introduction

Large language models (LLMs) (OpenAI, 2022; Thoppilan et al., 2022; Chowdhery et al., 2022) have brought revolutions in natural language processing (NLP) due to the remarkable zero-shot and few-shot generalization. One of the most well-known LLMs, ChatGPT (OpenAI, 2022) powered by GPT3.5 and GPT4 (OpenAI, 2023), has exhibited strong dialogue capabilities. As a closed model, ChatGPT sparked a lot of work for its evaluation and application on diverse tasks and aspects (Qin et al., 2023; Wei et al., 2023; Liang et al., 2023).

Information extraction (IE) is a fundamental topic in NLP, which aims to extract structured information from unstructured text, including tasks such as named entity recognition (NER) (Yu et al., 2020), relation extraction (RE) (Baldini Soares et al., 2019), event extraction (EE) (Chen et al., 2015), etc. Evaluating ChatGPT's performance on IE is important for understanding its capabilities in structured prediction and language understanding (Li et al., 2023; Wei et al., 2023).

With recent techniques for eliciting complex multi-step reasoning (Wei et al., 2022; Wang et al., 2022b), LLMs have shown remarkable zero-shot reasoning ability in arithmetic and symbolic reasoning (Kojima et al., 2022). However, the reasoning ability of LLM on IE remained unexplored. To mitigate this gap, we present a systematic empirical study exploring the reasoning capability of LLM on IE, with a focus on the ChatGPT and zero-shot NER task. By adapting the prevalent reasoning techniques (Zhou et al., 2022; Wei et al., 2022; Wang et al., 2022b) to NER, we propose the following strategies to stimulate the reasoning potential of LLM on NER:

- We break down the NER task into a series of simpler subproblems by labels and perform a **decomposed-question-answering (Decomposed-QA)** paradigm, where the model extracts entities of only one label at a time.

- We propose **syntactic augmentation** of two ways: **syntactic prompting**, which encourages the model to first analyze the syntactic structure of the input text itself, then recog-

---

*Corresponding authors.
[1]Code available at: https://github.com/Emma1066/Zero-Shot-NER-with-ChatGPT

nize the named entities based on the syntactic structure; **tool augmentation**, which provides the syntactic information generated by a parsing tool to the model.

- We tailor the self-consistency (SC) (Wang et al., 2022b) for NER and propose a **two-stage majority voting** strategy: after sampling multiple responses of the model, we first vote for the most consistent mentions, then the most consistent types.

The main contributions of this paper include:

- We present a systematic empirical investigation of zero-shot NER with LLM, with a specific emphasis on ChatGPT as one of the most robust LLMs available.

- We adapt prevalent reasoning methods to NER and propose four strategies tailored for NER: decomposed-QA, syntactic prompting, tool augmentation, and two-stage majority voting.

- We evaluate our strategies across seven benchmarks. Experiment results reveal that the proposed strategies significantly facilitate zero-shot NER across domain-specific out-of-distribution and general-domain datasets, including Chinese and English scenarios.

## 2 Related Work

### 2.1 Reasoning with LLM

LLM has shown remarkable zero-shot reasoning ability, in the way of explicitly encouraging the LLM to generate intermediate rational for solving a problem. On the one hand, recent works, in both the few-shot (Wei et al., 2022; Zhang et al., 2022; Wang et al., 2022a) and zero-shot (Kojima et al., 2022) setting, elicit chain-of-thought (CoT) from LLM and modify the answer by step-by-step. On the other hand, the problem decomposition, like least-to-most prompting (Zhou et al., 2022), reduces complex problems to the sub-problems, and then solves these sub-problems sequentially; the SC strategy (Wang et al., 2022b) generates a diverse set of answers by sampling from LLM, and then marginalizes out the sampled answers to determine the optimal answer. In this work, we focus on investigating the zero-shot reasoning ability of LLM on the NER task.

### 2.2 LLM on IE

A few works study the performance of the powerful LLM ChatGPT (Li et al., 2023; Ma et al., 2023; Laskar et al., 2023) on IE tasks. Wei et al. (2023) propose a two-stage chatting paradigm for IE. At stage one, it asks ChatGPT to recognize the types of elements; at stage two, it asks ChatGPT to extract the mentions corresponding to each type recognized at stage one. Han et al. (2023) presents an analysis of ChatGPT's performance on IE tasks from four aspects: performance, evaluation criteria, robustness, and errors. Wang et al. (2023) apply in-context learning (ICL) to NER by inserting special tokens into the demonstrations retrieved from the training set. Wan et al. (2023) apply CoT to relation extraction (RE) and use ChatGPT to generate intermediate rationales for demonstrations retrieved from the training set. Different from previous works, we focus on exploring the ChatGPT abilities for zero-shot reasoning on IE, with a focus on the NER task. We explore the prevalent reasoning methods with LLM, which exhibited remarkable performance on arithmetic and logical reasoning tasks. Most importantly, these methods are first adapted to the NER task based on the task characteristics.

## 3 Method

Adapting the prevalent reasoning techniques (Zhou et al., 2022; Wei et al., 2022; Wang et al., 2022b) to NER, we propose four strategies to stimulate the reasoning capabilities of LLM on NER. Examples of the proposed methods are shown in Fig. 1.

### 3.1 Decomposed-QA

Inspired by least-to-most prompting (Zhou et al., 2022), we improve zero-shot NER by decomposing the task into a set of simpler questions. Recognizing entities of all labels at one time may be too challenging for ChatGPT (as the vanilla zero-shot method shown in (a) of Fig. 1), especially when the label size is large, or the data is from a specific out-of-distribution domain. This motivates us to break down the NER task by labels. Given an input sentence, the whole process of recognizing entities is a multi-turn dialogue paradigm. Each time, ChatGPT is asked to recognize entities of a single label. After ChatGPT provides its response to the current question, we proceed to ask questions related to the next label, incorporating all the previous questions and answers as part of the dialogue context. Once all questions pertaining to each label have been addressed, we conclude the entire conversation.

We name this paradigm Decomposed-QA. The

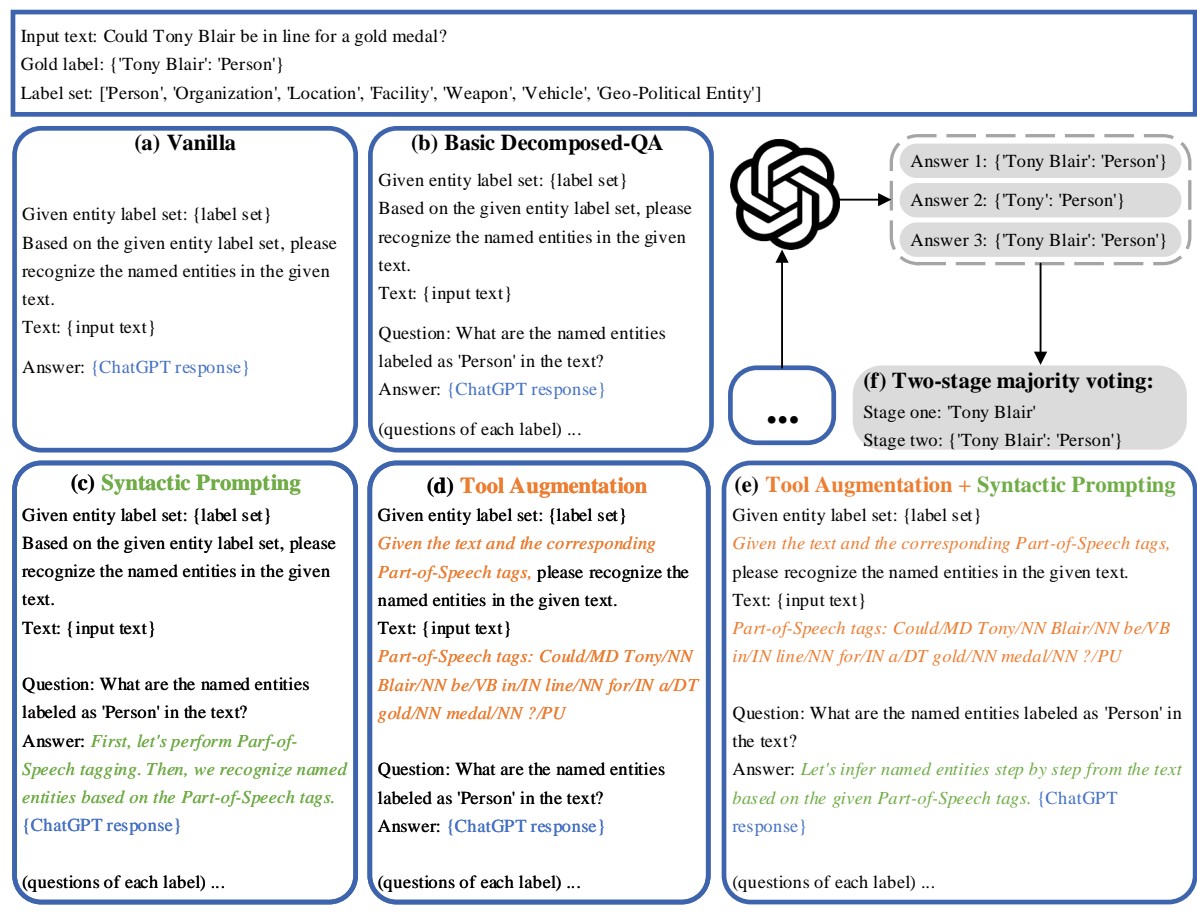

Figure 1: Examples of proposed methods for zero-shot NER with ChatGPT. (a) Vanilla zero-shot method. (b) Basic **decomposed-QA**, where the NER task is broken down into simpler subproblems. (c) Decomposed-QA with **syntactic prompting**. Texts in green are the proposed *syntactic reasoning hint* . (d) Decomposed-QA with **tool augmentation**. Texts in orange are the *content of syntactic information.* (e) Decomposed-QA with tool augmentation and syntactic prompting. (f) SC with **two-stage majority voting**, where stage one votes for the mentions and stage two votes for types. We use part-of-speech tags as an example syntactic information in this figure. The detailed prompts are shown in Appendix H.

example is shown in (b) of Fig. 1.

We obtain the label order used in the multi-turn dialogue by asking ChatGPT. For each dataset, we provide the task requirement and the label set to ChatGPT, then ask it to give a reasonable label order based on its understanding of the labels. For domain-specific datasets, PowerPlantFlat and PowerPlantNested, which will be introduced in Section 4.1, we also use a manual label order provided by the domain experts. The label orders are shown in Appendix G.

## 3.2 Syntactic Augmentation

Aiming to guide the model to think step by step while extracting information, we encourage Chat-GPT to first grasp the syntactic structure of the input text and then leverage this syntactic structure to extract relevant information. Among them,

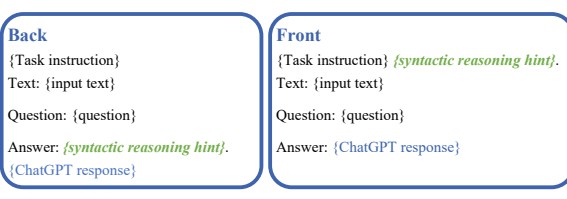

Figure 2: Two positions of syntactic reasoning hint.

five kinds of syntactic information are utilized: word segmentation, noun phrases, Part-of-Speech (POS) tags, constituency trees, and dependency trees. Word segmentation is only for Chinese. We propose the following two ways of syntactic augmentation.

**Syntactic Prompting**. We encourage the model to analyze the syntactic structure itself by inserting the *syntactic reasoning hint* in the input instruction, as shown in (c) of Fig. 1. We explore two positions

of syntactic reasoning hint, *i.e.*, in the back or front of the instruction, as shown in Fig. 2.

**Tool Augmentation**. We first obtain the syntactic information of the input text via a parsing tool;[2] Then, we feed the input text together with the syntactic information to ChatGPT, as shown in (d) of Fig. 1. We do not apply noun phrases in tool augmentation since we do not obtain a parsing tool with a reliable ability to extract noun phrases.

We further explore the combination of tool augmentation and syntactic prompting. To enhance the utilization of syntactic information from the parsing tool, we insert a syntactic reasoning hint. The example is shown in (e) of Fig. 1.

### 3.3 Self-Consistency with Two-Stage Majority Voting

Harnessing the power of SC (Wang et al., 2022b), we sample multiple responses from the model and select the most acknowledged answers as the final prediction. We design a two-stage majority voting for NER, as shown in (f) of Fig. 1. At stage one, for each candidate mention appeared in all responses, we consider it as an entity if it appeared in more than half of the responses; otherwise, we discard this mention. At stage two, for each mention kept in stage one, we choose the entity label predicted by the majority of responses as the final predicted label.

We explore two levels of SC for decomposed-QA: question-level and sample-level. For question-level, we sample multiple responses for the current question and conduct majority voting; then, we fill the voted answer into the dialogue context for all subsequent questions. For sample-level, we run the whole dialogue multiple times independently and obtain the answer of each run, then conduct majority voting on these answers.

## 4 Experiment

### 4.1 Setup

**Datasets.** We evaluate ChatGPT performance on both domain-specific and general-domain datasets. For domain-specific datasets, we present two Chinese NER datasets of the electric power domain, **PowerPlantFlat (PPF)** and **PowerPlantNested (PPN)**. The two datasets are collected from the

technical reports, which are formed during nuclear power plant operation and maintenance. PowerPlantFlat only contains flat cases, while PowerPlantNested contains nested entities. The two datasets are formed in the vertical industrial domain, and thus serve as out-of-distribution data for ChatGPT. The statistics of the two datasets are shown in Appendix A. For general-domain datasets, we evaluate on commonly used benchmarks, including two English datasets, ACE05,[3] and ACE04,[4] and three Chinese datasets, OntoNotes 4 (Onto. 4),[5] MSRA (Zhang et al., 2006) and Weibo NER (Peng and Dredze, 2015). For evaluation on more datasets, please refer to Appendix E.

**Model.** We mainly evaluate on GPT-3.5 (gpt-3.5-turbo) with official API.[6] For Decomposed-QA, we maintain a dialogue paradigm for each test sample. For vanilla setting, we generate the response separately for each test sample.

We also evaluate on GPT-3 (text-davinci-003) (Ouyang et al., 2022) and Llama2 (Touvron et al., 2023) to verify the effectiveness of the proposed methods on other LLMs. We use the 13B chat model of Llama2.[7] The results of these two LLMs are in Section 4.5.

**Self-consistency.** We set the temperature to 0.7 and 0 for settings with and without SC, respectively. For cost saving, we conduct majority voting of 5 responses in our main experiments. We first conduct both question-level and sample-level consistency on each dataset; then, we choose the way of higher performance for the rest of the experiments on the corresponding dataset.

**Data sampling.** For syntactic augmentation, we evaluate on the entire test sets of seven datasets. For SC and combinations of techniques, for cost saving, we evaluate on partial datasets and randomly sampled subsets of test sets: We evaluate on the two domain-specific datasets, PowerPlantFlat and PowerPlantNested, with entire test sets, and two general-domain datasets, Ontonotes 4 and ACE05, by randomly sampling 300 samples from the test set three times and reporting the average results.

---

[2]We use Hanlp (He and Choi, 2021) to generate syntactic information, since we found it performs well on both Chinese and English in our preliminary experiments.

[3]catalog.ldc.upenn.edu/LDC2006T06
[4]catalog.ldc.upenn.edu/LDC2005T09
[5]catalog.ldc.upenn.edu/LDC2011T03
[6]The results of ChatGPT are obtained during May and June 2023 with official API.
[7]https://huggingface.co/meta-llama/Llama-2-13b-chat-hf

| | Method | PPF | PPN | Weibo | MSRA | Onto. 4 | ACE05 | ACE04 |
|---|---|---|---|---|---|---|---|---|
| | Vanilla | 27.85 | 20.43 | 30.09 | 45.51 | 33.74 | 28.12 | 20.09 |
| | Decomposed-QA | **36.57** | **30.14** | **34.04** | **48.60** | **37.45** | 34.37 | **22.19** |
| Syn. / Front | Word segmentation | **38.16** | 30.38 | 32.72 | **47.52** | 37.47 | - | - |
| | Noun phrases | 37.46 | 30.02 | **33.93** | 46.05 | **38.31** | 33.22 | 20.99 |
| | POS tag | 36.89 | **30.60** | 32.68 | 46.87 | 36.82 | **34.31** | **21.74** |
| | Constituency tree | 36.21 | 29.88 | 31.85 | 46.02 | 36.52 | 33.22 | 20.86 |
| | Dependency tree | 36.33 | 29.82 | 33.49 | 45.61 | 35.90 | 34.21 | 21.04 |
| Syn. / Back | Word segmentation | 34.89 | 25.87 | 32.43 | **48.74** | 37.48 | - | - |
| | Noun phrases | 32.59 | 24.32 | 28.71 | 46.84 | 38.27 | **29.36** | 21.74 |
| | POS tag | **36.18** | **26.11** | **33.51** | 44.40 | 36.82 | 28.84 | 23.88 |
| | Constituency tree | 35.71 | 23.93 | 30.46 | 45.84 | **39.00** | 21.37 | 18.81 |
| | Dependency tree | 31.05 | 21.02 | 27.61 | 44.87 | 38.52 | 25.57 | 21.04 |
| Tool. | Word segmentation | 39.77 | 33.81 | 36.30 | 53.67 | 39.20 | - | - |
| | POS tag | 38.11 | 30.97 | 35.14 | 51.99 | 37.61 | **34.33** | **22.41** |
| | Constituency tree | 36.51 | 30.25 | 32.00 | 48.32 | 38.40 | 32.96 | 22.15 |
| | Dependency tree | 39.50 | 32.12 | 36.16 | 48.82 | 38.05 | 33.38 | 22.37 |
| | SOTA (fully-supervised) | 68.54 | 70.41 | 72.77 | 96.72 | 84.47 | 90.90 | 90.30 |

Table 1: Overall performance. We report the F1 values. **Vanilla** for vanilla zero-shot method without any techniques; **Syn.** for syntactic prompting; **Tool.** for tool augmentation. We use the same abbreviations in the rest of this paper when necessary. Syntactic augmentation is all conducted under the decomposed-QA setting. Numbers in **bold** are the best results in the corresponding categories; Numbers underlined are the best results among all methods in the zero-shot scenario. The proposed decomposed-QA and syntactic augmentation achieve significant improvements for zero-shot NER on both Chinese and English datasets and on both domain-specific and general-domain scenarios.

**SOTA of fully-supervised methods.** For PowerPlantFlat and PowerPlantNested, we use GlobalPointer (Su et al., 2022) since it performs well on both flat and nested cases. For other benchmarks, we refer to corresponding papers: Weibo (Wang et al., 2021), MSRA (Li et al., 2020), Ontonotes 4 (Li et al., 2020), ACE05 (Zhong and Chen, 2021), ACE04 (Zhong and Chen, 2021).

## 4.2 Overall Performance

Table 1 summarizes the performances of decomposed-QA and syntactic augmentation. For the two domain-specific datasets, we use the manual label orders as they show better performance in preliminary experiments. For cost saving, we explore SC and combinations of reasoning techniques on selected datasets and sampled test sets, which are detailed in Section 4.3

### 4.2.1 Effect of Decomposed-QA

From Table 1, we have the following observations: (1) Compared with the vanilla method, decomposed-QA achieves significant improvements across all benchmarks, including both Chinese and English scenarios, and both domain-specific and general-domain scenarios. This demonstrates that decomposing by labels makes the NER task much more manageable for ChatGPT. (2) Decomposed-QA exhibits more significant improvements on domain-specific datasets

(with an average 9.22% F1 gain) than on general-domain datasets (with an average 3.82% F1 gain). This is presumably because out-of-distribution data are more challenging for ChatGPT. Decomposing makes ChatGPT acquire a better understanding of the out-of-distribution data. (3) We also explore the effect of reasoning techniques under the vanilla setting, and the results are in Table 9 of Appendix C. We found that the vanilla setting fails to stimulate the potential of reasoning techniques. Contrarily, decomposed-QA stimulates the potential of syntactic augmentation.

### 4.2.2 Effect of Syntactic Augmentation

As shown in Table 1, we draw the following conclusions: (1) Syntactic prompting alone brings limited benefits. This is presumably because conducting syntactic analysis without any other augmentation is challenging for ChatGPT. (2) Tool augmentation exhibits consistent improvements across six datasets, showing that syntactic information helps ChatGPT better understand the input text. (3) Tool augmentation achieves more improvements on Chinese than English datasets. This may be due to the fact that Chinese is harder than English for ChatGPT to handle, and syntactic information provides a clue on how to understand the Chinese input better. (4) Different kinds of syntactic information exhibit various performances. On Chinese datasets, word segmentation shows the best performance.

| Method | | | PPF | PPN | Onto. 4 | ACE05 |
|---|---|---|---|---|---|---|
| | Vanilla | | 27.85 | 20.43 | 35.16 (1.57) | **29.45** (0.69) |
| | + SC | | **28.85** | **20.72** | **35.79** (1.36 ) | 29.37 (1.35 ) |
| Decomposed-QA | - | | **36.57** | 30.14 | 38.79 (1.66) | **35.57** (0.83) |
| + SC | question-level | | 33.46 | **32.15** | **39.57** (1.50) | 31.98 (0.31) |
| | sample-level | | 26.98 | 31.92 | 39.15 (0.76) | 34.38 (0.85) |
| Syn. | Front | Word segmentation | **38.16** | 30.38 | 37.67 (1.22) | - |
| | | Noun phrases | 37.46 | 30.02 | **38.83** (1.24) | 34.63 (0.78) |
| | | POS tag | 36.89 | **30.60** | 37.94 (1.49) | 34.28 (0.45) |
| | | Constituency tree | 36.21 | 29.88 | 38.43 (0.84) | 34.47 (0.77) |
| | | Dependency tree | 36.33 | 29.82 | 36.85 (1.16) | **35.77** (0.45) |
| | Back | Word segmentation | 34.89 | 25.87 | 39.16 (1.52) | - |
| | | Noun phrases | 32.59 | 24.32 | 39.52 (0.82) | **29.78** (0.64) |
| | | POS tag | **36.18** | **26.11** | 37.00 (2.41) | 29.72 (2.06) |
| | | on_conj | 35.71 | 23.93 | **40.53** (2.54) | 22.23 (0.40) |
| | | Dependency tree | 31.05 | 21.02 | 39.06 (2.88) | 26.65 (0.78) |
| Syn. + SC | Front | Word segmentation | **38.64** | **32.32** | 39.23 (1.13) | - |
| | | Noun phrases | 38.16 | 32.11 | **40.34** (1.30) | 32.35 (1.18) |
| | | POS tag | 38.06 | 31.75 | 38.71 (1.91) | 33.02 (1.11) |
| | | Constituency tree | 37.24 | 31.60 | 38.99 (1.52) | 32.00 (0.42) |
| | | Dependency tree | 37.65 | 31.30 | 37.17 (2.21) | **34.59** (0.14) |
| | Back | Word segmentation | 38.43 | 30.81 | 40.23 (2.59) | - |
| | | Noun phrases | **38.73** | 29.19 | 39.79 (2.24) | **34.92** (0.72) |
| | | POS tag | 38.48 | 30.77 | **40.27** (1.37) | 34.40 (1.93) |
| | | Constituency tree | 38.02 | **31.31** | 39.84 (1.90) | 33.95 (0.90) |
| | | Dependency tree | 37.24 | 31.20 | 40.15 (1.94) | 34.42 (0.37) |
| Tool. | | Word segmentation | **39.77** | **33.81** | 40.78 (2.58) | - |
| | | POS tag | 38.11 | 30.97 | 38.15 (2.82) | **35.35** (0.34) |
| | | Constituency tree | 36.51 | 30.25 | 38.54 (3.19) | 34.54 (2.26) |
| | | Dependency tree | 39.50 | 32.12 | 38.13 (3.04) | 34.34 (0.52) |
| Tool. + SC | | Word segmentation | 39.63 | 33.97 | **41.84** (2.63) | - |
| | | POS tag | 37.92 | 31.72 | 38.96 (4.21) | 33.42 (0.64) |
| | | Constituency tree | 36.59 | 28.35 | 40.40 (3.98) | **34.60** (0.21) |
| | | Dependency tree | **40.86** | 33.59 | 38.82 (2.61) | 30.69 (0.97) |
| Tool. + Syn. | Front | Word segmentation | **39.67** | **32.97** | 41.09 (3.19) | - |
| | | POS tag | 38.85 | 31.82 | 39.69 (3.98) | **36.78** (1.36) |
| | | Constituency tree | 36.02 | 30.65 | 39.44 (2.92) | 33.51 (3.04) |
| | | Dependency tree | 37.16 | 32.06 | 38.83 (3.29) | 34.09 (0.78) |
| | Back | Word segmentation | **36.24** | **31.46** | **39.68** (1.15) | - |
| | | POS tag | 34.71 | 26.51 | 36.62 (1.05) | **35.70** (1.17) |
| | | Constituency tree | 33.76 | 29.53 | 39.67 (1.55) | 29.64 (2.95) |
| | | Dependency tree | 33.18 | 27.73 | 36.85 (0.43) | 29.19 (2.17) |
| Tool. + Syn. + SC | Front | Word segmentation | **40.31** | **34.85** | **42.46** (2.20) | - |
| | | POS tag | 38.21 | 30.89 | 40.86 (2.48) | 33.19 (1.39) |
| | | Constituency tree | 35.76 | 29.00 | 41.36 (3.58) | **33.42** (2.35) |
| | | Dependency tree | 39.97 | 33.23 | 40.49 (3.49) | 30.29 (0.71) |
| | Back | Word segmentation | 40.83 | 30.78 | **41.40** (2.81) | - |
| | | POS tag | 38.00 | 30.64 | 38.58 (2.77) | **30.28** (2.21) |
| | | Constituency tree | 36.26 | 26.36 | 40.53 (3.38) | 29.78 (1.64) |
| | | Dependency tree | **41.97** | **32.73** | 40.19 (2.13) | 29.87 (0.17) |
| SOTA (fully-supervised) | | | 68.54 | 70.41 | 84.47 | 90.90 |

Table 2: Performance of SC and combinations of reasoning techniques. We report the F1 values. Numbers in parentheses are the standard deviations. Numbers in **bold** are the best results in the corresponding categories; Numbers underlined are the best results among all methods in the zero-shot scenario. SC with two-stage majority voting and combinations of reasoning techniques brings further improvements.

On English datasets, POS tags boost the most. This is presumably because simpler syntactic information is easier for ChatGPT to understand. Complex syntactic information, such as dependency tree, though informative, can be hard to understand, thereby, exhibiting unstable performance.

## 4.3 Effect of Self-Consistency and Combinations of Reasoning Techniques

Table 2 summarizes the performance of SC and the combinations of reasoning techniques. We visualize the results on PowerPlantFlat and Ontonotes 4

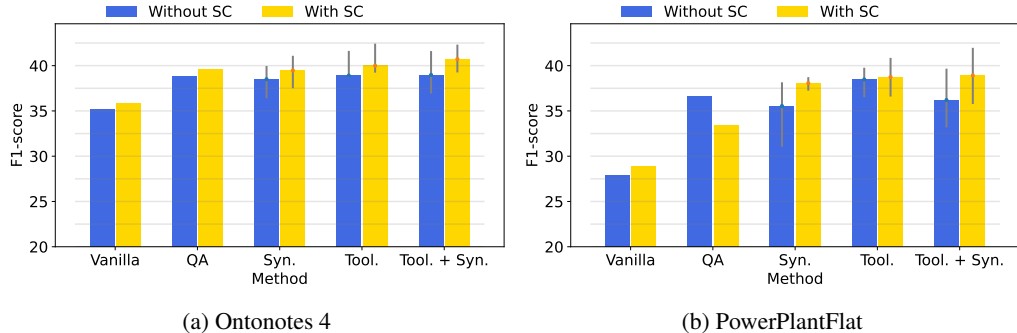

(a) Ontonotes 4        (b) PowerPlantFlat

Figure 3: Performance of combinations of reasoning techniques. For methods involving syntactic augmentation, we plot the average results over all kinds of syntactic information. The vertical lines on the top part of some bars represent the performances range over all kinds of syntactic information. With SC of two-stage majority voting, the combinations of reasoning techniques further improve the performances.

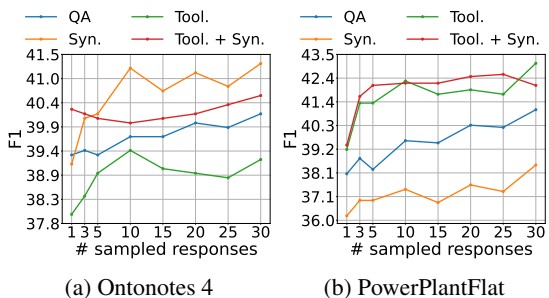

(a) Ontonotes 4      (b) PowerPlantFlat

Figure 4: Increasing sampled responses generally improves performance under SC with two-stage majority voting.

in Fig. 3 for better analysis.

From the table and the figure, we have the following observations and conclusions: (1) SC shows consistent improvements on almost all methods. As long as the syntactic information is involved, SC can always boost performance. This may be due to the fact that syntactic information is helpful but hard to understand or analyze. Thus, syntactic information gives ChatGPT the potential to perform better but also a higher possibility of making mistakes. SC can filter out errors, thereby, leveraging the advantages, and eliminating the disadvantages of syntactic information. (2) Syntactic prompting fails to boost tool augmentation and even hurts the performance. However, when equipped with SC, syntactic prompting improves tool augmentation. This may be due to the complexity of information provided by the combination of tool augmentation and syntactic prompting. The complex information leads the model to think and explore more, and of course, it is also accompanied by more possibilities for errors. This makes SC an effective means of filtering out errors here. (3) SC improves more when syntactic reasoning hints are put on the back than on the front. This is presumably because the closer

| Error Types | | Vanilla | QA | TS-SC |
|---|---|---|---|---|
| Type | OOD types | 4 | **1** | **1** |
| | Wrong types | **141** | 150 | 156 |
| Boundary | Cotain gold. | 70 | 54 | **35** |
| | Cotained by gold. | **9** | 27 | 24 |
| | Overlap with gold. | **0** | 1 | **0** |
| Completely-O | | 334 | 220 | **176** |
| Omitted mentions | | **23** | 41 | 43 |
| OOD mentions | | **3** | 36 | 10 |
| Total | | 585 | 530 | **444** |

Table 3: Numbers of error types on Ontonotes 4. "QA" for decomposed-QA, "**TS-SC**" for combinations of tool augmentation, syntactic prompting, and SC. Numbers in **bold** denote the best results, *i.e.*, the least errors. The proposed methods significantly reduce the total amount of error.

the reasoning hint is to the answer, the more it can stimulate the model's thinking. Hence, putting the reasoning hints on the back encourages the model to generate more diverse answers, which provides better search spaces for majority voting.

We explore the effect of increasing sampled responses in SC, which are shown in Fig. 4. We sample up to 30 responses for cost saving. As seen in the figure, sampling a higher number of responses improves the performance. We conjecture that combining diverse syntactic information may further benefit SC on NER.

## 4.4 Error Analysis

### 4.4.1 Error Types

We take Ontonotes 4 for error analysis. Table 3 summarizes the statistics of error types. Fig. 6 visualize the percentages of error types. Below is the introduction of error types:

**Type**. *OOD types*: predicted entity types not in the given label set; *Wrong types*: predicted types

Figure 5: Case study of error correction and error increase with the proposed methods. We translate the original Chinese text into English in the demonstrations for readability. The upper two cases are errors corrected, and the lower two are errors increased. Texts in blue are involved entities in the error cases. Our method shows effectiveness on error corrections. With the suggested optimization strategies, the error increased might be eliminated.

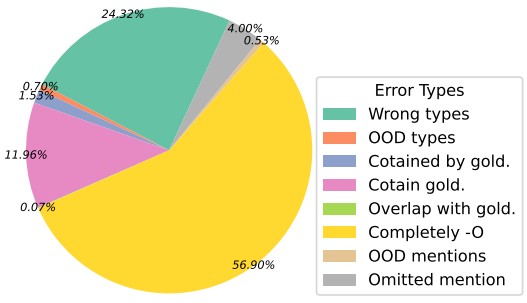

Figure 6: Percentage of different error types on Ontonotes 4 under the vanilla method.

incorrect but in the given label set.

**Boundary**. ***Contain gold.***: predicted mentions containing gold mentions; ***Contained by gold.***: predicted mentions contained by gold mentions; ***Overlap with gold.***: predicted mentions not in the two above situations but overlap with gold mentions.

***Completely-O***: predicted mentions that do not have any of the three above boundary situations with any gold mentions.

***OOD mentions***: predicted mentions that do not appear in the input text.

As shown in Fig. 6, the majority error types are *complete-O* and *wrong types*, which account for over 80% of all errors. The former may be due to

the incomplete annotation or that ChatGPT would guess entities based on its prior common knowledge. The latter may be due to the inadequate understanding of entity types. As seen in Table 3, decomposed-QA reduces the total error numbers by 9.4%; The combination of **T**ool augmentation, **S**yntactic prompting and **SC** (**TS-SC**) reduces the error numbers by 24.1%, showing remarkable capability in error corrections.

### 4.4.2 Case Study of Error Correction and Error Increase

As seen in Table 3, TS-SC reduces errors mainly in types of *contain gold.* and *completely-O*, and increases errors mainly in types of *contained by gold.* and *omitted mentions*. Thus, we conduct case study on these four types, which are shown in Fig. 5. TS-SC corrects errors of *contain gold.* and *completely-O* presumably by providing syntactic information and making the model better understand the input text. Meanwhile, TS-SC increases errors of *contain gold.* and *omitted mentions* presumably because of the misguiding of syntactic information and inadequate understanding of entity types, respectively. For the former, providing more accurate and comprehensive syntactic information might be a solution; for the latter, providing type information might be a direction of optimization.

| Dataset | Method | 0-shot | 3-shot | 5-shot | 10-shot |
|---|---|---|---|---|---|
| Ontonotes 4 | Vanilla | 35.16 (1.57) | 38.67 (3.57) | 44.51 (5.78) | 52.45 (4.13) |
| | Standard CoT | - | 34.34 (6.61) | 41.13 (6.31) | 41.90 (2.43) |
| | Tool. w. word segmentation (Ours) | **40.78** (2.58) | 42.48 (3.34) | 47.16 (5.42) | 54.40 (2.68) |
| | Syn. w. word segmentation (Ours) | 37.94 (1.49) | **43.89** (3.67) | **50.70** (7.26) | **56.71** (3.70) |
| PowerPlantFlat | Vanilla | 27.85 | 35.81 (2.94) | 37.44 (3.88) | 41.13 (4.89) |
| | Standard CoT | - | 30.63 (6.45) | 33.95 (3.59) | 38.02 (1.03) |
| | Tool. w. word segmentation (Ours) | **32.41** | **39.43** (1.91) | **41.12** (4.35) | 42.05 (4.74) |
| | Syn. w. word segmentation (Ours) | 28.09 | 37.84 (2.59) | 39.72 (2.79) | **42.52** (3.71) |

Table 4: Results under few-shot setting, where the number of shots is the number of texts. We randomly sample three sets of demonstrations and take the averages. Results for Ontonotes 4 are averaged over three sets of randomly sampled 300 samples from the test set. We report F1 values. Numbers in parentheses are the standard deviations. Numbers in **bold** are the best results. Our methods also achieve significant improvements in few-shot scenarios.

| Dataset | ACE05 | | | BC5CDR | | |
|---|---|---|---|---|---|---|
| Model | GPT-3.5 | GPT-3 | Llama2 | GPT-3.5 | GPT-3 | Llama2 |
| Vanilla | 29.45 | 14.03 | 9.07 | 61.28 | 29.49 | 26.12 |
| Decomposed-QA | **35.57** | 23.88 | 15.53 | **65.45** | 38.73 | 28.30 |
| Syn. w. dependency tree | 26.65 | **27.93** | 16.98 | 59.69 | 41.62 | 34.46 |
| Tool. w. dependency tree | 34.34 | 27.59 | 17.31 | 62.79 | **43.69** | **39.94** |
| Tool. + Syn. w. dependency tree | 29.19 | 18.38 | **26.99** | 57.28 | 16.38 | 39.57 |

Table 5: Performance on GPT-3 (text-davinci-003) and Llama2 13B chat model. Results are averaged over three sets of randomly sampled 300 samples from the test set. We report the F1 values. Our proposed strategies show consistent improvements on various LLMs.

## 4.5 More analysis

**Few-shot setting.** We evaluate the proposed syntactic augmentation under few-shot setting. Han et al. (2023) investigate standard CoT on NER by generating intermediate rationales with ChatGPT. We take a different perspective: we encourage the model to explore syntactic information as their intermediate thinking steps. Detailed adaptations of our methods to the few-shot setting are explained in Appendix D. For the decomposed-QA and SC, we leave them to future work due to the cost budget.

We compared our methods to the vanilla method and standard CoT. We use ChatGPT to generate rationales in standard CoT, following (Han et al., 2023). Here, we use one general domain dataset, Ontonotes 4, and one domain-specific dataset, PowerPlantFlat, for demonstrations. The results are shown in Table 4, in which the word segmentation is used for demonstration. The results of various syntactic information are in Appendix D.

As observed in Table 4, the standard CoT does not bring improvements, even hurt the performance. This is presumably because standard CoT is very sensitive to the rationales constructed, which is also mentioned in (Han et al., 2023). However, our strategies have achieved significant improvements. This shows that the proposed methods are effective not only in the zero-shot scenario but also in the few-shot setting.

**Other LLMs.** We also evaluate our methods on GPT-3 (text-davinci-003) (Ouyang et al., 2022) and Llama2 (Touvron et al., 2023). Since Llama2 still has poor support for Chinese yet, we evaluate on two English datasets, one general-domain dataset, ACE05, and one biomedical dataset, BC5CDR (Li et al., 2016). The results are shown in Table 5, in which the dependency tree is used for demonstration. The complete results are in Appendix F. The main results of BC5CDR are in Appendix E. Table 5 shows that our methods exhibit consistent improvements across different LLMs, including the close-sourced ChatGPT model series and typical open-sourced model Llama.

## 5 Conclusion

We present an empirical study of zero-shot NER with ChatGPT, with four proposed strategies to simulate the reasoning potential of ChatGPT on NER. Inspired by the powerful reasoning capabilities of LLM on logical and arithmetic reasoning tasks, the proposed strategies involve task decomposition, syntactic augmentation, and tailored SC. We verify the effectiveness of our methods on Chinese and English scenarios, and on both domain-specific and general-domain datasets. We provide an analysis of the error types with suggested solutions. Besides, we verify the effectiveness of the proposed methods on the few-shot setting and other LLMs.

## 6  Limitations

For cost saving, we focus on the investigation of each individual syntactic information and have not explored the combinations of different kinds of syntactic information. Also, we have not investigated manual label orders on general-domain datasets for the same reason. We leave them to future work.

## Acknowledgments

We would like to thank the anonymous reviewers for their insightful comments and constructive suggestions. This research is supported by the National Key Research and Development Program of China (Grant No. 2020YFB1707803) and Zhejiang Provincial Natural Science Foundation of China (LDT23F02023F02).

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

# A  Statistics of PowerPlant Datasets

Table 6 shows the overall statistics of the PowerPlant datasets, and Table 8 displays the classwise statistics.

| Dataset | Split | #Sentences | #Entities |
|---------|-------|------------|-----------|
| Flat | Train | 3087 | 4379 |
| | Test | 401 | 540 |
| Nested | Train | 3047 | 6924 |
| | Test | 492 | 1109 |

Table 6: Statistics of PowerPlant datasets.

# B  Statistics of Errors on PowerPlantFlat

Table 7 summarizes the statistics of error types on PowerPlantFlat under vanilla, decomposed-QA, and TS-SC methods. Fig. 7 shows the percentages of error types on PowerPlantFlat under vanilla method.

| Error Types | | Vanilla | QA | TS-SC |
|-------------|--|---------|-----|-------|
| Type | OOD types | **0** | **0** | **0** |
| | Wrong types | 90 | 66 | **65** |
| Boundary | Cotain gold. | 153 | 167 | **110** |
| | Cotained by gold. | **31** | 51 | 76 |
| | Overlap with gold. | 8 | 8 | **3** |
| Completely-O | | 439 | 406 | **279** |
| Omitted mentions | | 65 | **37** | 56 |
| OOD mentions | | **14** | 32 | **14** |
| Total | | 800 | 767 | **603** |

Table 7: Numbers of different error types on PowerPlantFlat. "**QA**" refers to decomposed-QA, "**TS-SC**" refers to the combination of tool augmentation, syntactic prompting, and SC. Numbers in bold denote the best results on PowerPlantFlat, i.e., the least errors.

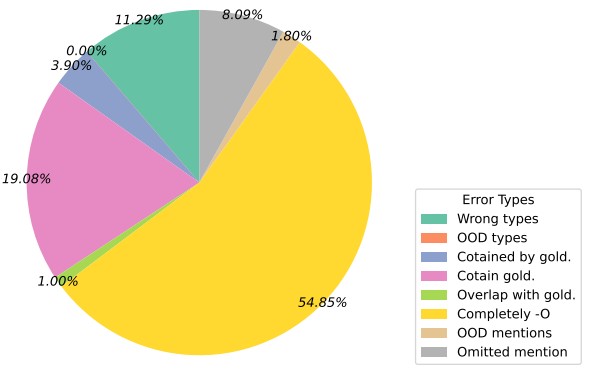

Figure 7: Percentage of different error types on PowerPlantFlat under the vanilla setting.

## C Performance Under Vanilla Setting

We also investigate the effect of our proposed reasoning techniques on the standard setting. The results are shown in Table 9. From the table, we can conclude that the potential of the syntactic information cannot be fully exploited under the standard setting. On the contrary, the proposed decomposed-QA paradigm effectively utilizes the syntactic information, as shown in Tabel 2 and Figure 3.

Under standard setting, the reasoning techniques bring limited benefits for general-domain datasets, sometimes even hurting the performance. However, these techniques exhibit improvements on domain-specific datasets, *i.e.*, out-of-distribution datasets. This is presumably because out-of-distribution data is much more challenging than general-domain data for ChatGPT. The reasoning techniques lead the model to have a better understanding of the out-of-distribution data.

## D Syntactic Augmentation Under Few-shot Setting

The following are the adaptations of our proposed syntactic augmentation strategies to the few-shot setting. (1) Syntactic prompting: For the test sample, we ask the model to first perform syntactic analysis and then recognize entities. For demonstrations, we use parsing tools to generate intermediate syntactic parsing results. (2) Tool augmentation: We provide both the text and syntactic information for the demonstrations and the test sample.

Table 10 shows the experiment results under the 3-shot setting.

## E Evaluation on More Datasets

We additionally evaluate the proposed methods on more datasets, including general-domain datasets, CoNLL-2003 (Sang and De Meulder, 2003), WNUT-17(Derczynski et al., 2017), and domain-specific datasets (i.e., biomedical domain), BC5CDR (Li et al., 2016), BioNLP11 (Pyysalo et al., 2012), and CRAFT (Bada et al., 2012; Crichton et al., 2017).

The results are shown in Table 11. We found that the proposed reasoning techniques cannot guarantee performance improvements on CoNLL-2003 and WNUT-17 and even hurt the performance. This is presumably because the label logic of these two datasets is not suitable for decomposition, and the syntactic information generated for them is noisy.

Meanwhile, we conjecture that this is also due to the fact that CoNLL-2003 and WNUT-17 contain more numbers of shorter texts, on which the reasoning techniques are difficult to leverage their advantages. However, the proposed methods achieve significant improvements in biomedical domain datasets BC5CDR, BioNLP11, and CRAFT. This demonstrates that the proposed methods can also improve zero-shot NER of other challenging domains besides the electric power domain. Plus the five datasets evaluated in Table 11, we evaluate on **twelve** benchmarks in total and achieve remarkable improvements on **ten** datasets among them.

## F Evaluation on Other LLMs

The complete results on GPT-3 and Llama2 are shown in Table 12. These results show that our methods exhibit consistent improvements across different LLMs. On the smallest LLM evaluated, Llama2 13B, our proposed strategies still achieve remarkable performance improvements, with 19.72% and 17.51% F1 improvements on ACE05 and BC5CDR, respectively. This reveals that our methods have wide applicability to various sizes of LLMs, which is beneficial for low-resource scenarios such as when only smaller LLMs are affordable.

## G Label Order

Table 13 displays label orders used in our main experiments and the corresponding results under basic decomposed-QA. On the power plant datasets, manual label orders provided by domain experts achieve significantly better results. This demonstrates that when dealing with domain-specific datasets with ChatGPT, one may turn to domain knowledge to boost performance.

Table 14 displays the label orders of additional datasets.

Table 15 shows the instructions for asking Chat-GPT to provide label orders of PowerPlantFlat and ACE05 datasets.

## H Prompts

We show all of our prompts with Ontonotes 4 and ACE05 as examples. The prompts are in Table 16, 17, 18 and 19.

| Label | Chinese Label | Flat | | Nested | |
|---|---|---|---|---|---|
| | | Train | Test | Train | Test |
| System name | 系统名称 | 132 | 10 | 143 | 21 |
| System identity | 系统标识 | 357 | 49 | 1654 | 270 |
| Device name | 设备名称 | 1239 | 159 | 1191 | 199 |
| Device identity | 设备标识 | 1517 | 185 | 1462 | 243 |
| Component name | 部件名称 | 763 | 97 | 771 | 108 |
| Location name | 地点 | 200 | 24 | 197 | 30 |
| Person | 人员 | 171 | 16 | 184 | 24 |
| Reactor Status | 反应堆状态 | - | - | 88 | 13 |
| Power Plant Event | 电站事件 | - | - | 1234 | 201 |

Table 8: Classwise statistics of PowerPlant datasets.

| Method | | PPF | PPN | Weibo | MSRA | Ontonotes 4 | ACE04 | ACE05 |
|---|---|---|---|---|---|---|---|---|
| Vanilla | | 27.85 | 20.43 | 30.09 | 45.51 | 33.74 | 20.09 | 28.12 |
| Self-Consistency | | 28.85 | 20.72 | 31.02 | - | - | 19.97 | 28.21 |
| Syntactic Prompting | | | | | | | | |
| Front | Word segmentation | 28.09 | 20.37 | 28.48 | 41.72 | 30.82 | - | - |
| | Noun phrases | 28.94 | 21.81 | 28.89 | 41.5 | 30.89 | 18.77 | 26.21 |
| | POS tags | 30.12 | 22.47 | 27.23 | 41.20 | 30.59 | 19.54 | 28.27 |
| | Constituency Tree | 26.38 | 20.47 | 28.23 | 40.62 | 30.61 | 19.75 | 28.49 |
| | Dependency Tree | 27.21 | 20.7 | 28.51 | 40.44 | 30.77 | 19.68 | 28.46 |
| Back | Word segmentation | 27.37 | 20.58 | 20.10 | 42.92 | 32.03 | - | - |
| | Noun phrases | 31.65 | 21.36 | 17.09 | 42.60 | 31.62 | 19.69 | 26.05 |
| | POS tags | 28.24 | 17.70 | 19.38 | 42.34 | 31.86 | 20.60 | 25.65 |
| | Constituency Tree | 30.31 | 20.53 | 17.74 | 42.52 | 31.88 | 20.42 | 23.98 |
| | Dependency Tree | 26.08 | 17.47 | 14.09 | 42.68 | 31.64 | 20.33 | 26.31 |
| Tool augmentation | | | | | | | | |
| Front | Word segmentation | 32.28 | 26.57 | 25.37 | 38.84 | 29.75 | - | - |
| | POS tags | 28.13 | 24.17 | 24.86 | 37.29 | 29.95 | 19.14 | 28.14 |
| | Constituency Tree | 23.62 | 20.97 | 22.98 | 30.45 | 26.1 | 16.97 | 27.65 |
| | Dependency Tree | 26.08 | 17.47 | 14.09 | 42.68 | 31.64 | 20.33 | 26.31 |
| Back | Word segmentation | 28.57 | 26.81 | 21.88 | 34.19 | 26.77 | - | - |
| | POS tags | 22.04 | 202.5 | 24.69 | 34.77 | 27.84 | 18.16 | 25.17 |
| | Constituency Tree | 22.46 | 21.82 | 20.79 | 30.81 | 24.62 | 16.09 | 23.60 |
| | Dependency Tree | 21.36 | 20.25 | 25.18 | 32.73 | 26.88 | 16.13 | 21.67 |
| SOTA (fully-supervised) | | 68.54 | 70.41 | 72.77 | 96.72 | 84.47 | 90.3 | 90.9 |

Table 9: Performance of reasoning techniques under the vanilla setting (without decomposition). In this table, "**vanilla**" specifically refers to the zero-shot method without any techniques. We report the F1 values on entire test sets. We spare the SC on MSRA and Ontonotes 4 for cost saving.

| Method | | Ontonotes 4 | PowerPlantFlat |
|---|---|---|---|
| Vanilla | | 38.71 (3.34) | 35.81 (2.94) |
| Decomposed-QA | | **43.30** (1.84) | **43.75** (3.06) |
| Syn. | Word segmentation | 40.12 (3.22) | **37.33** (2.70) |
| | POS tag | **44.11** (3.52) | 36.42 (0.26) |
| | Constituency tree | 38.81 (2.38) | 33.66 (2.50) |
| | Dependency tree | 35.05 (1.41) | 36.16 (1.43) |
| Syn. + SC | Word segmentation | 41.28 (3.65) | 38.7 (2.27) |
| | POS tag | **44.93** (4.02) | 36.19 (0.65) |
| | Constituency tree | 41.89 (3.50) | 36.05 (2.65) |
| | Dependency tree | 37.98 (2.56) | **39.06** (0.78) |
| Tool. | Word segmentation | 42.33 (3.14) | **39.43** (1.91) |
| | POS tag | **42.51** (2.44) | 37.04 (0.72) |
| | Constituency tree | 38.51 (4.17) | 35.14 (2.84) |
| | Dependency tree | 36.12 (1.93) | 33.54 (1.44) |
| Tool. + SC | Word segmentation | 40.49 (12.05) | **41.09** (2.71) |
| | POS tag | **43.28** (2.69) | 38.66 (2.05) |
| | Constituency tree | 40.12 (4.31) | 35.54 (2.66) |
| | Dependency tree | 38.39 (2.50) | 35.79 (1.87) |
| Tool. + Syn. | Word segmentation | **43.11** (2.52) | **39.69** (2.61) |
| | POS tag | 42.44 (2.67) | 36.35 (1.81) |
| | Constituency tree | 38.31 (3.30) | 34.45 (2.53) |
| | Dependency tree | 35.21 (2.26) | 34.1 (1.06) |
| Tool. + Syn. + SC | Word segmentation | **42.18** (2.29) | **41.43** (1.78) |
| | POS tag | 41.53 (3.25) | 37.95 (2.03) |
| | Constituency tree | 41.57 (4.54) | 35.32 (3.82) |
| | Dependency tree | 40.33 (4.87) | 35.85 (2.14) |

Table 10: Performance of syntactic augmentation under 3-shot setting. We randomly sample three sets of demonstrations and report the means and standard deviations of F1 values. Numbers in parentheses are standard deviations. Numbers in **bold** are the best results in each category. The proposed syntactic augmentation exhibits significant improvements in the few-setting.

| Dataset | CoNLL-2003 | WNUT-17 | BC5CDR | BioNLP11 | CRAFT |
|---|---|---|---|---|---|
| Vanilla | **69.42** (0.91) | **46.61** (2.97) | 61.28 (3.11) | 51.29 (2.48) | 21.66 (1.41) |
| Decomposed-QA | 59.67 (0.36) | 42.39 (1.99) | **65.45** (0.89) | **55.3** (0.54) | **23.99** (2.65) |
| Syn. (Front) | | | | | |
| Noun phrases | **57.13** (0.41) | **39.75** (2.42) | 64.41 (1.74) | 52.73 (1.89) | 22.92 (3.04) |
| POS tag | 55.14 (0.98) | 39.74 (1.98) | **66.24** (2.40) | 53.97 (0.99) | **23.59** (2.01) |
| Constituency tree | 56.36 (1.07) | 39.66 (1.51) | 64.7 (0.80) | 53.98 (1.24) | 23.84 (1.94) |
| Dependency tree | 54.27 (1.54) | 38.36 (0.37) | 65.56 (1.20) | **54.37** (0.83) | 23.51 (2.06) |
| Syn. (Back) | | | | | |
| Noun phrases | **58.89** (1.39) | 36.47 (1.10) | **60.44** (0.57) | 51.99 (1.27) | 21.71 (1.89) |
| POS tag | 56.12 (1.54) | **38.12** (0.55) | 58.19 (1.55) | 54.41 (2.62) | 22.69 (3.76) |
| Constituency tree | 55.66 (2.17) | 37.75 (2.04) | 47.81 (1.99) | 43.58 (1.44) | 23.86 (1.30) |
| Dependency tree | 58.36 (1.11) | 37.49 (1.56) | 59.69 (0.76) | **55.23** (0.77) | **22.84** (1.59) |
| Tool. | | | | | |
| POS tag | **62.79** (2.54) | 43.81 (2.15) | **66.4** (1.44) | **52.38** (0.35) | 24.54 (3.87) |
| Constituency tree | 60.96 (0.34) | **44.3** (1.02) | 65.02 (3.00) | 51.44 (0.81) | 23.88 (2.24) |
| Dependency tree | 59.23 (3.13) | 41.6 (1.30) | 62.79 (2.62) | 42.71 (0.73) | **24.86** (2.45) |
| Tool. + Syn. (Front) | | | | | |
| POS tag | **63.46** (1.02) | 43.9 (2.21) | 64.24 (1.83) | 49.87 (0.85) | **25.05** (2.12) |
| Constituency tree | 59.59 (0.99) | **44.68** (2.59) | **65.43** (2.51) | **50.6** (1.39) | 24.5 (2.76) |
| Dependency tree | 57.93 (2.05) | 40.96 (3.57) | 59.93 (1.88) | 40.18 (1.63) | 24.46 (2.50) |
| Tool. + Syn. (Back) | | | | | |
| POS tag | 58.08 (0.27) | **39.32** (4.42) | **60.84** (3.47) | **47.9** (1.88) | 13.71 (1.90) |
| Constituency tree | **58.95** (2.19) | 37.91 (5.46) | 54.57 (1.86) | 42.27 (2.78) | 20.64 (1.85) |
| Dependency tree | 54.68 (1.27) | 36.7 (3.07) | 57.28 (1.37) | 45.62 (3.46) | **24.05** (4.05) |

Table 11: Performance on additional datasets. Results are averaged over three sets of randomly sampled 300 samples from the test set. We report the means and standard deviations of F1 values. Numbers in parentheses are standard deviations. Numbers in **bold** are best results in each category.

| Dataset | ACE05 | | | BC5CDR | | |
|---|---|---|---|---|---|---|
| Model | GPT-3.5 | GPT-3 | Llama2 | GPT-3.5 | GPT-3 | Llama2 |
| Vanilla | 29.45 (0.69) | 14.03 (0.94) | 9.07 (1.33) | 61.28 (3.11) | 29.49 (3.12) | 26.12 (2.94) |
| Decomposed-QA | **35.57** (0.83) | **23.88** (2.21) | **15.53** (1.53) | **65.45**(0.89) | **38.73** (2.58) | **28.30** (0.73) |
| Syn. (Front) | | | | | | |
| Noun phrases | 34.63 (0.78) | 21.74 (2.16) | 17.30 (1.08) | 64.41 (1.74) | 39.38 (3.01) | 36.20 (1.24) |
| POS tag | 34.28 (0.45) | 21.98 (2.90) | 23.86 (6.79) | **66.24** (2.40) | **45.31** (1.34) | 34.65 (2.35) |
| Constituency tree | 34.47 (0.77) | 22.38 (2.09) | 21.76 (0.31) | 64.70 (0.80) | 42.09 (1.51) | **36.92** (0.37) |
| Dependency tree | **35.77** (0.45) | **22.84** (2.81) | **25.91** (1.20) | 65.56 (1.20) | 43.19 (0.93) | 33.11 (0.31) |
| Syn. (Back) | | | | | | |
| Noun phrases | **29.78** (0.64) | 24.45 (2.29) | 15.73 (1.93) | **60.44** (0.57) | 35.17 (1.88) | 33.75 (1.26) |
| POS tag | 29.72 (2.06) | **30.73** (2.90) | 16.51 (1.82) | 58.19 (1.55) | **45.17** (3.00) | 34.62 (3.02) |
| Constituency tree | 22.23 (0.40) | 27.08 (2.82) | 16.45 (1.74) | 47.81 (1.99) | 39.72 (1.96) | **35.17** (0.88) |
| Dependency tree | 26.65 (0.78) | 27.93 (2.73) | **16.98** (1.23) | 59.69 (0.76) | 41.62 (2.30) | 34.46 (1.81) |
| Tool. | | | | | | |
| POS tag | **35.35** (0.34) | 24.74 (0.88) | **18.00** (1.02) | **66.40** (1.44) | 47.04 (2.39) | **40.45** (1.11) |
| Constituency tree | 34.54 (2.26) | 26.84 (2.46) | 17.36 (0.65) | 65.02 (3.00) | **52.77** (2.73) | 38.95 (0.85) |
| Dependency tree | 34.34 (0.52) | **27.59** (2.05) | 17.31 (2.14) | 62.79 (2.62) | 43.69 (2.33) | 39.94 (1.05) |
| Tool. + Syn. (Front) | | | | | | |
| POS tag | **36.78** (1.36) | 26.21 (1.88) | 17.94 (1.28) | 64.24 (1.83) | 47.70 (2.46) | 33.84 (1.63) |
| Constituency tree | 33.51 (3.04) | 29.93 (2.03) | **18.11** (1.42) | **65.43** (2.51) | **54.12** (3.00) | 30.23 (1.65) |
| Dependency tree | 34.09 (0.78) | **30.62** (1.79) | 15.75 (2.38) | 59.93 (1.88) | 43.26 (2.11) | **38.16** (4.50) |
| Tool. + Syn. (Back) | | | | | | |
| POS tag | **35.70** (1.17) | **22.08** (1.32) | 24.50 (2.09) | **60.84** (3.47) | **17.72** (1.87) | **43.63** (2.61) |
| Constituency tree | 29.64 (2.95) | 15.06 (0.23) | 23.87 (1.18) | 54.57 (1.86) | 11.44 (1.72) | 36.48 (1.91) |
| Dependency tree | 29.19 (2.17) | 18.38 (1.10) | **26.99** (0.49) | 57.28 (1.37) | 16.38 (1.27) | 39.57 (2.04) |

Table 12: Complete results on various LLMs. We use gpt-3.5-turbo for GPT-3.5, text-davinci-003 for GPT-3, and 13B chat model for Llama2. For cost saving, we sample 300 samples from the test set three times, and report the average results of F1 values. Numbers in parentheses are the standard deviations. Numbers in **bold** are the best results in the corresponding categories.

| Dataset | Label order | Order generation | F1 |
|---|---|---|---|
| PowerPlantFlat | vanilla | - | 27.85 |
| | [["设备标识"],["设备名称"], ["系统标识"], ["系统名称"],["部件名称"],["地点"],["人员"]] | manual | 36.57 |
| | [["地点"],["系统名称"], "系统标识"], ["设备名称"], ["设备标识"], ["部件名称"], ["人员"]] | ChatGPT | 30.52 |
| PowerPlantNested | vanilla | - | 20.43 |
| | [["设备标识"],["设备名称"], ["系统标识"], ["系统名称"],["部件名称"],["地点"], ["人员"], ["反应堆状态"],["电站事件"]] | manual | 30.14 |
| | [["地点"], ["人员"],["反应堆状态"], ["系统名称"], ["系统标识"], ["设备名称"], ["设备标识"], ["部件名称"], ["电站事件"]] | ChatGPT | 20.16 |
| Weibo | vanilla | - | 30.09 |
| | [['人名'], ['地名'], ['机构名称'], ['地缘政治实体']] | ChatGPT | 34.04 |
| MSRA | vanilla | - | 45.51 |
| | [['人物'], ['地点'], ['机构']] | ChatGPT | 48.60 |
| Ontonotes 4 | vanilla | - | 33.74 |
| | [['人名'], ['地名'], ['机构名称'], ['地缘政治实体']] | ChatGPT | 37.45 |
| ACE04 | vanilla | - | 20.09 |
| | [["Person"],["Organization"],["Location"], ["Facility"], ["Weapon"],["Vehicle"], ["Geo-Political Entity"]] | ChatGPT | 22.19 |
| ACE05 | vanilla | - | 28.12 |
| | [["Person"],["Organization"],["Location"], ["Facility"], ["Weapon"],["Vehicle"], ["Geo-Political Entity"]] | ChatGPT | 34.37 |

Table 13: Label orders with corresponding performances. The results are from the entire test set. "vanilla" refers to the standard setting without any techniques.

| Dataset | Label order | Order generation | F1 |
|---|---|---|---|
| CoNLL-2003 | vanilla | - | 69.42 |
| | [["Location"], ["Organization"], ["Person"], ["Miscellaneous"]] | ChatGPT | 59.67 |
| WNUT-17 | vanilla | - | 46.61 |
| | [["Person"], ["Location"], ["Corporation"], ["Product"], ["Creative work"], ["Group"]] | ChatGPT | 42.39 |
| BC5CDR | vanilla | - | 61.28 |
| | [["Chemical"], ["Disease"]] | ChatGPT | 65.45 |
| BioNLP11 | vanilla | - | 51.29 |
| | [['Protein'], ['Organism'], ['Chemical'], ['Regulon-operon']] | ChatGPT | 55.30 |
| CRAFT | vanilla | - | 21.66 |
| | [['Simple_chemical'], ['Gene_or_gene_product'], ['Cellular_component'], ['Complex']] | ChatGPT | 23.99 |

Table 14: Label orders of additional datasets and corresponding performances. Results are averaged over three sets of randomly sampled 300 samples from the test set.

| Dataset | Prompts |
|---|---|
| PowerPlantFlat | "给定实体标签集：['系统名称', '系统标识', '设备名称', '设备标识', '部件名称', '地点', '人员'] \n 我们需要按照不同标签分别识别对应的命名实体。按什么样的标签顺序是合理的？" |
| ACE05 | "Given entity label set: ['Person', 'Organization', 'Location', 'Facility', 'Weapon', 'Vehicle', 'Geo-Political Entity'] \n We need to recognize the corresponding named entities based on different labels. What is the reasonable label order?" |

Table 15: Instructions for asking ChatGPT to provide label orders.

| Syntactic prompting | |
|---|---|
| 给定实体标签集：['地缘政治实体'，'机构名称'，'地名'，'人名']\n 请基于给定的实体标签集，识别给定文本中的命名实体。syntactic reasoning hint (front) \n文本：中国保险监管项目在京启动\n

问题：文本中标签为'人名'的实体有哪些？请以如下JSON格式提供答案：[{'实体名称'：'实体标签'}]。如果没有对应实体，请返回如下空列表：[]。\n答案：{syntactic reasoning hint (back)}

问题：文本中标签为'地名'的实体有哪些？请以如下JSON格式提供答案：[{'实体名称'：'实体标签'}]。如果没有对应实体，请返回如下空列表：[]。\n答案：{syntactic reasoning hint (back)}

问题：文本中标签为'机构名称'的实体有哪些？请以如下JSON格式提供答案：[{'实体名称'：'实体标签'}]。如果没有对应实体，请返回如下空列表：[]。\n答案：{syntactic reasoning hint (back)}

问题：文本中标签为'地缘政治实体'的实体有哪些？请以如下JSON格式提供答案：[{'实体名称'：'实体标签'}]。如果没有对应实体，请返回如下空列表：[]。\n答案：{syntactic reasoning hint (back)} | |

| Syntactic reasoning hint (front) | |
|---|---|
| Word segmentation | 首先，你应该进行分词。接着，你应该基于分词结果识别命名实体。 |
| Noun phrases | 首先，你应该识别名词。接着，你应该基于名词识别命名实体。 |
| POS tagging | 首先，你应该进行词性标注。接着，你应该基于标注的词性识别命名实体。 |
| Constituency parsing | 首先，你应该进行成分句法解析。接着，你应该基于成分树识别命名实体。 |
| Dependency parsing | 首先，你应该进行依存句法解析。接着，你应该基于依存树识别命名实体。 |

| Syntactic reasoning hint (back) | |
|---|---|
| Word segmentation | 首先，让我们进行分词。接着，我们基于分词结果识别命名实 体。 |
| Noun phrases | 首先，让我们识别名词。接着，我们基于名词识别命名实体。 |
| POS tagging | 首先，让我们进行词性标注。接着，我们基于标注的词性识别命名实体。 |
| Constituency parsing | 首先，让我们进行成分句法解析。接着，我们基于成分树识别 命名实体。 |
| Dependency parsing | 首先，让我们进行依存句法解析。接着，我们基于依存树识别命名实体。 |

Table 16: Syntactic prompting on Ontonotes 4.

| Tool augmentation + syntactic prompting |
| --- |
| 给定实体标签集：['地缘政治实体', '机构名称', '地名', '人名']\n{task instruction (involving syntactic tool)}{syntactic reasoning hint (front)}\n文本：中国保险监管项目在京启动\n{syntactic information from tool} |
| 问题：文本中标签为'人名'的实体有哪些？请以如下JSON格式提供答案：[{'实体名称'：'实体标签'}]。如果没有对应实体，请返回如下空列表：[]。\n答案：{syntactic reasoning hint (back)} |

 (questions of each label) ...

| {Task instruction (involving syntactic tool)} | |
| --- | --- |
| Word segmentation | 给定文本和对应的分词结果，请基于实体标签集识别文本中的命名实体。 |
| POS tagging | 给定文本和对应的词性标注，请基于实体标签集识别文本中的命名实体。 |
| Constituency parsing | 给定文本和对应的成分树，请基于实体标签集识别文本中的命名实体。 |
| Dependency parsing | 给定文本和对应的依存树，请基于实体标签集识别文本中的命名实体。 |

| {Syntactic information from tool} | |
| --- | --- |
| Word segmentation | 分词：['中国', '保险', '监管', '项目', '在', '京', '启动']\n |
| POS tagging | 词性标注：中国/NR 保险/NN 监管/NN 项目/NN 在/P 京/NR 启动/VV\n |
| Constituency parsing | 成分树：(TOP\n (IP\n (NP (NP (NR 中国)) (NP (NN 保险) (NN 监管) (NN 项目)))\n (VP (PP (P 在) (NP (NR 京))) (VP (VV 启动)))))\n |
| Dependency parsing | 依存树：[['中国', '项目', 'nn'], ['保险', '项目', 'nn'], ['监管', '项目', 'nn'], ['项目', '启动', 'nsubj'], ['在', '启动', 'prep'], ['京', '在', 'pobj'], ['启动', '启动', 'root']]\n |

| {syntactic reasoning hint (front)} | |
| --- | --- |
| Word segmentation | 请基于给定的分词结果，从文本一步步推理出命名实体。 |
| POS tagging | 请基于给定的词性标注，从文本一步步推理出命名实体。 |
| Constituency parsing | 请基于给定的成分树，从文本一步步推理出命名实体。 |
| Dependency parsing | 请基于给定的依存树，从文本一步步推理出命名实体。 |

| {syntactic reasoning hint (back)} | |
| --- | --- |
| Word segmentation | 让我们基于给定的分词结果，从文本一步步推理出命名实体。 |
| POS tagging | 让我们基于给定的词性标注，从文本一步步推理出命名实体。 |
| Constituency parsing | 让我们基于给定的成分树，从文本一步步推理出命名实体。 |
| Dependency parsing | 让我们基于给定的依存树，从文本一步步推理出命名实体。 |

Table 17: Tool augmentation w. / wo. syntactic prompting on Ontonotes 4. If using syntactic prompting, fill in {syntactic reasoning hint}; If not, discard {syntactic reasoning hint}.

| Syntactic prompting |
|---|

Given entity label set: ['Person', 'Organization', 'Location', 'Facility', 'Weapon', 'Vehicle', 'Geo-Political Entity']\n{task instruction (involving syntactic tool)}{syntactic reasoning hint (front)} \nText: Could Tony Blair be in line for a gold medal?\n{syntactic information from tool}",

Question: What are the named entities labeled as 'Person' in the text? Provide the answer in the following JSON format: [{'Entity Name': 'Entity Label'}]. If there is no corresponding entity, return the following empty list: []. \nAnswer: {syntactic reasoning hint (back)}

(questions of each label) ...

| Syntactic reasoning hint (front) | |
|---|---|
| Noun phrases | First, you should recognize the noun phrases. Then, you should recognize named entities based on the noun phrases. |
| POS tagging | First, you should perform Part-of-Speech tagging. Then, you should recognize named entities based on the Part-of-Speech tags. |
| Constituency parsing | First, you should perform constituency parsing. Then, you should recognize named entities based on the constituency tree. |
| Dependency parsing | First, you should perform dependency parsing. Then, you should recognize named entities based on the dependency tree. |
| Syntactic reasoning hint (back) | |
| Noun phrases | First, let's recognize the noun phrases. Then, we recognize named entities based on the noun phrases. |
| POS tagging | First, let's perform Part-of-Speech tagging. Then, we recognize named entities based on the Part-of-Speech tags. |
| Constituency parsing | First, let's perform constituency parsing. Then, we recognize named entities based on the constituency tree. |
| Dependency parsing | First, let's perform dependency parsing. Then, we recognize named entities based on the dependency tree. |

Table 18: Syntactic prompting on ACE05.

| Tool augmentation + syntactic prompting |
| --- |

Given entity label set: ['Person', 'Organization', 'Location', 'Facility', 'Weapon', 'Vehicle', 'Geo-Political Entity']\n{task instruction (involving syntactic tool)}{syntactic reasoning hint (front)} \nText: Could Tony Blair be in line for a gold medal?\n{syntactic information from tool}",

Question: What are the named entities labeled as 'Person' in the text? Provide the answer in the following JSON format: [{'Entity Name': 'Entity Label'}]. If there is no corresponding entity, return the following empty list: []. \nAnswer: {syntactic reasoning hint (back)}

(questions of each label) ...

| {Task instruction (involving syntactic tool)} | |
| --- | --- |
| POS tagging | Given the text and the corresponding Part-of-Speech tags, please recognize the named entities in the given text. |
| Constituency parsing | Given the text and the corresponding constituency tree, please recognize the named entities in the given text. |
| Dependency parsing | Given the text and the corresponding dependency tree, please recognize the named entities in the given text. |

| {Syntactic information from tool} | |
| --- | --- |
| POS tagging | Part-of-Speech tags: Could/JJ Tony/NN Blair/NN be/NN in/P line/NN for/P a/CD gold/NN medal/NN ?/PU\n |
| Constituency parsing | Constituency tree: (TOP\n (NP\n (NP\n (NP (ADJP (JJ Could)) (NP (NN Tony) (NN Blair) (NP (NN be))))\n (PP (P in) (NP (NN line))))\n (PP (P for) (NP (QP (CD a)) (NP (NN gold) (NN medal))))\n |
| Dependency parsing | Dependency tree: [['Could', 'be', 'amod'], ['Tony', 'be', 'nn'], ['Blair', 'be', 'nn'], ['be', '?', 'root'], ['in', 'be', 'prep'], ['line', 'in', 'pobj'], ['for', 'be', 'prep'], ['a', 'medal', 'nummod'], ['gold', 'medal', 'nn'], ['medal', 'for', 'pobj'], ['?', 'be', 'punct']]\n |

| {syntactic reasoning hint (front)} | |
| --- | --- |
| POS tagging | Please infer named entities step by step from the text based on the given Part-of-Speech tags. |
| Constituency parsing | Please infer named entities step by step from the text based on the given constituency tree. |
| Dependency parsing | Please infer named entities step by step from the text based on the given dependency tree. |

| {syntactic reasoning hint (back)} | |
| --- | --- |
| POS tagging | Let's infer named entities step by step from the text based on the given Part-of-Speech tags. |
| Constituency parsing | Let's infer named entities step by step from the text based on the given constituency tree. |
| Dependency parsing | Let's infer named entities step by step from the text based on the given dependency tree. |

Table 19: Tool augmentation w. / wo. syntactic promptings on ACE05. If using syntactic prompting, fill in {syntactic reasoning hint}; If not, discard {syntactic reasoning hint}.