# OpenReview forum: "Empirical Study of Zero-Shot NER with ChatGPT"
_EMNLP/2023/Conference — EMNLP 2023 Main_

### Official Review · Reviewer_5LaW · 2023-08-05

**Soundness:** 3

**Excitement:**

4: Strong: This paper deepens the understanding of some phenomenon or lowers the barriers to an existing research direction.

**Paper Topic And Main Contributions:**

This paper explores the ability of an LLM (ChatGPT-3.5-Turbo) in zero-shot NER with additional syntactic information. The study incorporates a variety of ways to inform syntactic information in the prompt to ChatGPT 3.5 for performing the task of named entity recognition (NER). The experimental results show that the NER can be improved by providing syntactic information such as word segmentation (for Chinese) and POS tagging, while both constituency parsing and dependency parsing help relatively limited.
This is an interesting work showing the powerful LLM can still benefit from the information produced by the "traditional" NLP parsers.

The goal of this work is clear and well-motivated. The writing is easy to follow. A variety of settings were conducted in experiments.
However, only one LLM (ChatGPT-3.5-Turbo) and one syntactic parser (Hanlp) were included. As a result, the effects of individual LLM and parser are not clear. More LLMs and parsers can be added for comprehension. In addition, is an even better result reachable when the golden-ground (human labeled) word-segmentation and POS tagging are fed?

**Questions For The Authors:**

Experimental results show a large gap between the zero-shot LLM and the state-of-the-art fully supervised model, showing that the significant limitation of LLMs in NER. However, LLMs have been shown capable in many complex tasks, do you think the ability in NER still important?

**Reasons To Accept:**

* This work shows the LLM can still be improved by the information produced by "traditional" NLP parsers.
* A variety of syntactic information and settings are evaluated in the experiments.
* Experimental results lead to the conclusions that are useful for the NLP community.

**Reasons To Reject:**

* Only one LLM (ChatGPT-3.5-Turbo) and one syntactic parser (Hanlp) are included in the experiments; the effects of individual LLM and parser are not clear.

**Reproducibility:**

4: Could mostly reproduce the results, but there may be some variation because of sample variance or minor variations in their interpretation of the protocol or method.

**Reviewer Confidence:**

4: Quite sure. I tried to check the important points carefully. It's unlikely, though conceivable, that I missed something that should affect my ratings.

**Typos Grammar Style And Presentation Improvements:**

* Figure 5 is unreadable for non Chinese speakers.

* Putting footnote numbers after comma, period, and semi-colon.

---

> ### Author Rebuttal · Authors · 2023-08-29
>
> Thank you for your constructive comments. The major comments are answered below.
>
> **Question: Importance of LLMs ability in NER.**
>
> First, the zero-shot ability of LLMs could facilitate low-resource information extraction (IE). IE and knowledge construction are fundamental research fields with broad applications [1]. If the powerful zero-shot ability of LLMs can be extended to IE, we can achieve efficient IE and knowledge construction with extremely low resources.
>
> Second, knowledge bases might improve the ability of LLMs. LLMs still encounter hallucinations or lack of knowledge when solving knowledge-intensive tasks [2]. With the assistance of up-to-date and scenario-specific knowledge bases, LLMs might provide more reliable applications. If LLMs can achieve efficient knowledge extraction on its own and use the extracted knowledge to assist it in handling other tasks, the problem of hallucinations might be significantly alleviated.
>
> Therefore, we believe that the ability of LLMs in NER is important and worth exploring.
>
> [1] Zou X. A survey on application of knowledge graph[C]//Journal of Physics: Conference Series. IOP Publishing, 2020, 1487(1): 012016.
>
> [2] Yao S, Zhao J, Yu D, et al. React: Synergizing reasoning and acting in language models. ICLR, 2023.
>
> **Reasons To Reject: The effects of individual LLM and parser.**
>
> (a) Other LLMs.
>
> To achieve similar complex reasoning and instruction comprehension abilities, the model's size and training techniques are generally required to be similar to ChatGPT. However, we currently cannot afford to run such a large model locally. So, we focus on the evaluation of ChatGPT. As a powerful but closed-source model, the assessment of ChatGPT is important for this community. And we leave the evaluation of other LLMs to our future work.
>
> (b) Other syntactic parsers.
>
> Here, we evaluate another parsing tool, stanza, from Stanford University. To exclude the influence of other reasoning techniques and only observe the initial effects of different parsing tools, we conduct tool augmentation on four types of syntactic information under the vanilla paradigm. The results are shown in the table below.
>
> | Dataset (Language) | ACE05 (English) |              | PowerPlantFlat (Chinese) |        |
> |--------------------|-----------------|--------------|--------------------------|--------|
> | Parseing tool      | Hanlp           | Stanza       | Hanlp                    | Stanza |
> | Word segmentation  | \               | \            | **31.76**                    | 28.55  |
> | POS tag            | 30.71 (1.43)    | **31.51** (0.97) | **25.91**                    | 20.6   |
> | Constituency tree  | 28.06 (2.03)    | **28.64** (1.08) | **21.39**                    | 19.76  |
> | Dependency tree    | 28.32 (1.70)    | **28.38** (1.76) | **23.52**                   | 20.97  |
>
>
> The results show that the quality of parsing tools and their help in reasoning are related to the language used. Stanza, from English developers, has higher support for the English dataset, ACE05; Hanlp, from Chinese developers, has better support for the Chinese dataset, PowerPlantFlat. Therefore, when using our methods, we can choose parsing tools that are more suitable for the corresponding language.
>
> **Typos Grammar Style And Presentation Improvements.**
>
> Thank you for kindly pointing out the typo error. We will fix it.

---

### Official Review · Reviewer_B3Rk · 2023-08-05

**Soundness:** 4

**Excitement:**

3: Ambivalent: It has merits (e.g., it reports state-of-the-art results, the idea is nice), but there are key weaknesses (e.g., it describes incremental work), and it can significantly benefit from another round of revision. However, I won't object to accepting it if my co-reviewers champion it.

**Paper Topic And Main Contributions:**

The paper introduces a new approach for zero-shot named entity recognition (NER) using ChatGPT. The proposed approach contains two key components: decomposed question-answering paradigm and syntactic augmentation. The decomposed question-answering paradigm breaks down the NER task into simpler subproblems by labels. The syntactic augmentation method consists of two aspects: syntactic prompting and tool augmentation. The paper also adopts the self-consistency method to enhance performance. The paper conducts extensive experiments on various English and Chinese datasets. The experimental results demonstrate the effectiveness of the proposed approach.

**Questions For The Authors:**

Did you explore applying this approach to GPT-4 yet? GPT-4 usually outperforms ChatGPT by a considerable margin.

**Reasons To Accept:**

1. The paper is clear and easy to understand. The figures are very clear. The appendices are also thoroughly detailed.
2. The paper proposes a novel approach utilizing LLMs for NER and conducts extensive experiments. The paper also conducts a detailed error analysis. This could assist future research focused on the application of LLMs for NER.
3. The paper conducts experiments on both Chinese and English datasets.

**Reasons To Reject:**

1. Efficiency concerns. The proposed decomposed question-answering paradigm breaks down the NER task into simpler subproblems by labels. Therefore, the approach requires multiple inquiries to ChatGPT. For one example, the number of inquiries is proportional to the number of entity types in the schema. Such efficiency is relatively low, which may affect the real-world application of this algorithm.
2. Experimental results. The performance of the proposed approach is much lower than the previous SoTA. Although the comparison may not be entirely fair, as the SoTA employs a fully supervised methodology whereas this approach is zero-shot, the performance of this approach is still quite a distance away from being practically usable. I wonder if the model's performance could potentially be improved by using few-shot prompting.

**Reproducibility:**

4: Could mostly reproduce the results, but there may be some variation because of sample variance or minor variations in their interpretation of the protocol or method.

**Reviewer Confidence:**

4: Quite sure. I tried to check the important points carefully. It's unlikely, though conceivable, that I missed something that should affect my ratings.

---

> ### Author Rebuttal · Authors · 2023-08-29
>
> Thank you for your constructive comments. We hope these could address your concerns.
>
> **Question: Exploration of applying the approaches to GPT-4.**
>
> Due to the price being 20-30 times more expensive than GPT-3.5 (20 times for input and 30 times for output), we currently cannot afford the extensive use of GPT-4. We tested some individual examples and found that GPT-4 has excellent reasoning and comprehension abilities. Our method might achieve a more significant improvement on GPT-4. We will verify it in our future work.
>
> **Reasons To Reject: Efficiency concerns.**
>
> Using label combinations in decomposed-QA could significantly reduce the rounds of inquiries. Combining similar/related labels in one round of inquiry can improve the efficiency. Due to time and budget constraints, we only explored one-by-one label orders in this work. We will explore label combinations according to label relations and logic in our future work. Here, we conduct preliminary experiments. Below is the used one-by-one label order and label group order of Ontonotes 4 and PowerPlantFlat. We show them all in English for readability.
> * Ontonotes 4:
>   * One-by-one order: \[[Person], [Location], [Organization], [Geo-political entity]]
>   * Label group order: \[[Person, Organization], [Geo-political entity, Location]]
> * PowerPlantFlat:
>   * One-by-one order: \[[Device Identity], [Device Name], [System Identity], [System Name], [Component Name], [Location], [Person]]
>   * Label group order: \[[Device Identity, Device Name], [System Identity, System Name], [Component Name], [Location], [Person]]
>
> The results below reveal that appropriate label combinations can also achieve superior performance in decomposed-QA paradigm.
>
> | Dataset           | Ontonotes 4  | PowerPlantFlat |
> |-------------------|--------------|----------------|
> | Vanilla           | 35.16 (1.57) | 27.85          |
> | QA - one by one   | 38.79 (1.66) | 36.57          |
> | QA - label groups | 40.01 (1.58) | 36.37          |
>
>
>
> **Reasons To Reject: Experimental results.**
>
> (a) Comparison to fully-supervised SOTA.
>
> The goal of this work is to explore the performance boundary of zero-shot NER with ChatGPT. With various tailored reasoning techniques, our results show that the zero-shot results still need to catch up with fully-supervised SOTA. We consider this conclusion also contributes to the community.
>
> (b) Improvements of few-shot prompting.
>
> We conduct preliminary experiments on few-shot settings. We encourage the model to explore syntactic information as their intermediate thinking. The following are the adaptations of our proposed zero-shot strategies to the few-shot setting.
> * Decomposed-QA: We provide the complete dialogue of each demonstration and let the model complete the dialogue of the test sample.
> * Syntactic prompting: For the test sample, we ask the model to first perform syntactic analysis and then recognize entities. For demonstrations, we use parsing tools to generate intermediate syntactic parsing results.
> * Tool augmentation: We provide both the text and the syntactic information for the demonstrations and test sample.
> * Self-consistency with two-stage majority voting: the method is the same as in the zero-shot setting.
>
> The following table shows the experiment results under the **3-shot** setting. We randomly sample three sets of demonstrations and report the means and standard deviations.
>
> |                            | Ontonotes 4  | PowerPlantFlat  |
> |----------------------------|---------------|-----------------|
> | Vanilla                    | 38.71 (3.34)  | 35.81 (2.94)    |
> | Decomposed-QA              | **43.30** (1.84)   | **43.75**(3.06)    |
> | Syn.(front)                |               |                 |
> | Word segmentation          | 40.12 (3.22)  | **37.33** (2.70)    |
> | POS tag                    | **44.11** (3.52)  | 36.42 (0.26)    |
> | Constituency tree          | 38.81 (2.38)  | 33.66 (2.50)    |
> | Dependency tree            | 35.05 (1.41)  | 36.16 (1.43)    |
> | Syn. + Consistency         |               |                 |
> | Word segmentation          | 41.28 (3.65)  | 38.7 (2.27)     |
> | POS tag                    | **44.93** (4.02)  | 36.19 (0.65)    |
> | Constituency tree          | 41.89 (3.50)  | 36.05 (2.65)    |
> | Dependency tree            | 37.98 (2.56)  | **39.06** (0.78)    |
> | Tool.                      |               |                 |
> | Word segmentation          | 42.33 (3.14)  | **39.43** (1.91)    |
> | POS tag                    | **42.51** (2.44)  | 37.04 (0.72)    |
> | Constituency tree          | 38.51 (4.17)  | 35.14 (2.84)    |
> | Dependency tree            | 36.12 (1.93)  | 33.54 (1.44)    |
> | Tool. + Consistency        |               |                 |
> | Word segmentation          | 40.49 (12.05) | **41.09** (2.71)    |
> | POS tag                    | **43.28** (2.69)  | 38.66 (2.05)    |
> | Constituency tree          | 40.12 (4.31)  | 35.54 (2.66)    |
> | Dependency tree            | 38.39 (2.50)  | 35.79 (1.87)    |
> | Tool. + Syn.               |               |                 |
> | Word segmentation          | **43.11** (2.52)  | **39.69** (2.61)    |
> | POS tag                    | 42.44 (2.67)  | 36.35 (1.81)    |
> | Constituency tree          | 38.31 (3.30)  | 34.45 (2.53)    |
> | Dependency tree            | 35.21 (2.26)  | 34.1 (1.06)     |
> | Tool. + Syn. + Consistency |               |                 |
> | Word segmentation          | **42.18** (2.29)  | **41.43** (1.78)    |
> | POS tag                    | 41.53 (3.25)  | 37.95 (2.03)    |
> | Constituency tree          | 41.57 (4.54)  | 35.32 (3.82)    |
> | Dependency tree            | 40.33 (4.87)  | 35.85 (2.14)    |
>
> For each dataset, we choose the two most effective syntactic augmentation methods and evaluate them on different numbers of demonstrations. The results are shown below. Due to time constraints, for combinations of decomposed-QA, syntactic augmentation, and self-consistency under few-shot settings, we leave them as our future work.
>
> | # Demonstrations                 | 0            | 3            | 5            | 10            |
> |---------------------------|--------------|--------------|--------------|---------------|
> | Ontonotes 4               |              |              |              |               |
> | Vanilla                   | 35.16 (1.57) | 38.67 (3.57) | 44.51 (5.78) | 52.45 (4.13)  |
> | Tool. (word segmentation) | **40.78** (2.58) | 42.48 (3.34) | 47.16 (5.42) | 54.40 (2.68)  |
> | Syn. (word segmentation)  | 37.94 (1.49) | **43.89** (3.67) | **50.70** (7.26) | **56.71** (3.70)  |
> | PowerPlantFlat            |              |              |              |               |
> | Vanilla                   | 27.85        | 35.81 (2.94) | 37.44 (3.88) | 41.13 (4.89)  |
> | Tool. (word segmentation) | **32.41**        | **39.43** (1.91) | **41.12** (4.35) | 42.05 (4.74)  |
> | Syn. (POS tag)            | 28.09        | 37.84 (2.59) | 39.72 (2.79) | **42.52** (3.71)  |
>
> The results in the above two tables show that our proposed strategies also achieve significant improvements under few-shot settings.

---

### Official Review · Reviewer_5Z6K · 2023-08-11

**Soundness:** 3

**Excitement:**

4: Strong: This paper deepens the understanding of some phenomenon or lowers the barriers to an existing research direction.

**Missing References:**

Try adding: Han, R., Peng, T., Yang, C., Wang, B., Liu, L., & Wan, X. (2023). Is Information Extraction Solved by ChatGPT? An Analysis of Performance, Evaluation Criteria, Robustness and Errors. ArXiv, abs/2305.14450. to your reference.

**Paper Topic And Main Contributions:**

This paper presents an empirical study of zero-shot named entity recognition (NER) using the large language model ChatGPT. The authors adapt common reasoning techniques used with LLMs for logical and arithmetic reasoning to the NER task. Their main contributions are proposing four strategies tailored for NER to elicit reasoning from ChatGPT: task decomposition by entity labels, syntactic augmentation, self-consistency, and two-stage majority voting. Experiments on English and Chinese benchmarks, including domain-specific datasets, demonstrate that their proposed techniques significantly improve zero-shot NER performance across domains. The analysis of error types provides insights into optimization directions. Overall, this is a thorough investigation of reasoning abilities for zero-shot NER using an available robust LLM. The adapted reasoning strategies are novel in facilitating multi-step inference for NER. The empirical results have important implications for low-resource NER scenario using LLM.

**Questions For The Authors:**

Question A:
The authors mention chain-of-thought (CoT) prompting but do not compare against standard CoT techniques. Could the authors include experiments on conventional CoT methods to better analyze the improvements from the reasoning-based approaches?

Question B:
For a more comprehensive assessment, could the authors evaluate their methods on additional widely-used NER datasets like CoNLL-2003 and WNUT-17? This would provide more context on the generalization capability.

Question C:
The authors used ChatGPT in the experiments. How do the authors think their techniques will transfer to other LLMs? Could the authors provide any preliminary analysis on another large model to demonstrate wider applicability?

Question D:
The results show significant improvements over the vanilla method, but there is still a large gap compared to supervised state-of-the-art. What are some potential ways to further enhance the reasoning and reduce this gap?

Question E:
The authors identified some optimization directions based on the error analysis. Could the authors elaborate on their next steps to address these limitations and refine the reasoning techniques?

**Reasons To Accept:**

The authors have comprehensively explored decomposition and augmentation techniques to enhance the zero-shot NER capabilities of large language models. Their proposed strategies, including label-wise task decomposition, syntactic augmentations, and self-consistency mechanisms, are shown to significantly boost NER performance across both English and Chinese datasets spanning different domains. The experiments demonstrate the efficacy of adapting common reasoning techniques in facilitating multi-step inference for zero-shot NER using the ChatGPT model. This empirical study provides valuable insights and guidance for low-resource NER settings where leveraging the reasoning and generalization abilities of large pre-trained models is critical. The authors have conducted thorough experiments to validate their proposed reasoning-based approaches for improving zero-shot NER.

**Reasons To Reject:**

The augmentation strategies proposed lack significant novelty - both the decomposed and syntactic augmentation approaches are basic and commonly used techniques in NLP tasks. The paper mentions chain-of-thought (CoT) but does not include experiments on standard CoT methods or typical in-context learning approaches with LLMs. Additionally, the authors did not evaluate on some commonly used NER datasets like CoNLL-2003 and WNUT-17. While the empirical study is solid, the enhancements explored are not particularly innovative. Comparing against standard CoT and few-shot learning strategies on more NER benchmarks would provide a more comprehensive assessment and additional context for the claims.

**Reproducibility:**

4: Could mostly reproduce the results, but there may be some variation because of sample variance or minor variations in their interpretation of the protocol or method.

**Reviewer Confidence:**

4: Quite sure. I tried to check the important points carefully. It's unlikely, though conceivable, that I missed something that should affect my ratings.

---

> ### Author Rebuttal · Authors · 2023-08-29
>
> We thank the reviewer for the comments and feedback. We hope these could address your concerns.
>
> **Question A: Not including standard chain-of-thought (CoT).**
>
> We conduct preliminary experiments to evaluate the proposed strategies under few-shot settings. Here, one general domain dataset, Ontonotes 4, and one domain-specific dataset, PowerPlantFlat, are employed for demonstrations.
>
> Work [1] investigates few-shot CoT on NER by generating intermediate rationales with ChatGPT. We take a different perspective: we encourage the model to explore syntactic information as their intermediate thinking. The following are the adaptations of our proposed zero-shot strategies to the few-shot setting.
> * Decomposed-QA: We provide the complete dialogue of each demonstration and let the model complete the dialogue of the test sample.
> * Syntactic prompting: For the test sample, we ask the model to first perform syntactic analysis and then recognize entities. For demonstrations, we use parsing tools to generate intermediate syntactic parsing results.
> * Tool augmentation: We provide both the text and the syntactic information for the demonstrations and test sample.
> * Self-consistency with two-stage majority voting: the method is the same as the zero-shot setting.
>
> The table below shows the experiment results under the **3-shot** setting. We randomly sample three sets of demonstrations and report the means and standard deviations.
>
> |                            | Ontonotes 4  | PowerPlantFlat  |
> |----------------------------|---------------|-----------------|
> | Vanilla                    | 38.71 (3.34)  | 35.81 (2.94)    |
> | Decomposed-QA              | **43.30** (1.84)   | **43.75**(3.06)    |
> | Syn.(front)                |               |                 |
> | Word segmentation          | 40.12 (3.22)  | **37.33** (2.70)    |
> | POS tag                    | **44.11** (3.52)  | 36.42 (0.26)    |
> | Constituency tree          | 38.81 (2.38)  | 33.66 (2.50)    |
> | Dependency tree            | 35.05 (1.41)  | 36.16 (1.43)    |
> | Syn. + Consistency         |               |                 |
> | Word segmentation          | 41.28 (3.65)  | 38.7 (2.27)     |
> | POS tag                    | **44.93** (4.02)  | 36.19 (0.65)    |
> | Constituency tree          | 41.89 (3.50)  | 36.05 (2.65)    |
> | Dependency tree            | 37.98 (2.56)  | **39.06** (0.78)    |
> | Tool.                      |               |                 |
> | Word segmentation          | 42.33 (3.14)  | **39.43** (1.91)    |
> | POS tag                    | **42.51** (2.44)  | 37.04 (0.72)    |
> | Constituency tree          | 38.51 (4.17)  | 35.14 (2.84)    |
> | Dependency tree            | 36.12 (1.93)  | 33.54 (1.44)    |
> | Tool. + Consistency        |               |                 |
> | Word segmentation          | 40.49 (12.05) | **41.09** (2.71)    |
> | POS tag                    | **43.28** (2.69)  | 38.66 (2.05)    |
> | Constituency tree          | 40.12 (4.31)  | 35.54 (2.66)    |
> | Dependency tree            | 38.39 (2.50)  | 35.79 (1.87)    |
> | Tool. + Syn.               |               |                 |
> | Word segmentation          | **43.11** (2.52)  | **39.69** (2.61)    |
> | POS tag                    | 42.44 (2.67)  | 36.35 (1.81)    |
> | Constituency tree          | 38.31 (3.30)  | 34.45 (2.53)    |
> | Dependency tree            | 35.21 (2.26)  | 34.1 (1.06)     |
> | Tool. + Syn. + Consistency |               |                 |
> | Word segmentation          | **42.18** (2.29)  | **41.43** (1.78)    |
> | POS tag                    | 41.53 (3.25)  | 37.95 (2.03)    |
> | Constituency tree          | 41.57 (4.54)  | 35.32 (3.82)    |
> | Dependency tree            | 40.33 (4.87)  | 35.85 (2.14)    |
>
> For each dataset, we choose the two most effective syntactic augmentation methods and evaluate them on different numbers of demonstrations. The results are shown below. Due to time constraints, for combinations of decomposed-QA, syntactic augmentation, and self-consistency under few-shot settings, we leave them as our future work.
>
> | # Demonstrations                 | 0            | 3            | 5            | 10            |
> |---------------------------|--------------|--------------|--------------|---------------|
> | Ontonotes 4               |              |              |              |               |
> | Vanilla                   | 35.16 (1.57) | 38.67 (3.57) | 44.51 (5.78) | 52.45 (4.13)  |
> | Tool. (word segmentation) | **40.78** (2.58) | 42.48 (3.34) | 47.16 (5.42) | 54.40 (2.68)  |
> | Syn. (word segmentation)  | 37.94 (1.49) | **43.89** (3.67) | **50.70** (7.26) | **56.71** (3.70)  |
> | PowerPlantFlat            |              |              |              |               |
> | Vanilla                   | 27.85        | 35.81 (2.94) | 37.44 (3.88) | 41.13 (4.89)  |
> | Tool. (word segmentation) | **32.41**        | **39.43** (1.91) | **41.12** (4.35) | 42.05 (4.74)  |
> | Syn. (POS tag)            | 28.09        | 37.84 (2.59) | 39.72 (2.79) | **42.52** (3.71)  |
>
> The results in the above two tables show that our proposed strategies also achieve significant improvements under few-shot settings.
>
> Last but not least, we emphasize that the goal of this work is to explore **zero-shot** NER with ChatGPT’s reasoning abilities. Standard CoT techniques [2] belong to **few-shot** learning strategies, which are **not** the main focus of our work.
>
> [1] Han, R., Peng, T., Yang, C., Wang, B., Liu, L., & Wan, X. (2023). Is Information Extraction Solved by ChatGPT? An Analysis of Performance, Evaluation Criteria, Robustness and Errors. ArXiv, abs/2305.14450.
>
> [2] Wei J, Wang X, Schuurmans D, et al. Chain-of-thought prompting elicits reasoning in large language models[J]. Advances in Neural Information Processing Systems, 2022, 35: 24824-24837.
>
> **Question B: More comprehensive assessment.**
>
> Below are the results on more datasets, including general-domain and domain-specific datasets (i.e., biomedical domain). The syntactic augmentation and two-stage voting are conducted under decomposed-QA paradigm.
>
> * We found that the proposed reasoning techniques cannot guarantee performance improvements on CoNLL-2003 and WNUT-16 and even hurt the performance. This is presumably because the label logic of these two datasets is not suitable for decomposition, and the syntactic information generated for them is noisy.
> * However, the proposed methods achieve significant improvements in biomedical domain datasets BC5CDR [1], BioNLP11 [2], and CRAFT [3] [4]. This demonstrates that the proposed methods can also improve zero-shot NER of other challenging specific domains besides the electric power domain evaluated in our paper.
> * Plus the **seven** datasets evaluated in our paper (Tables 1 and 2 of our paper), we have achieved significant improvements on **ten** datasets in total, including Chinese and English, on both domain-specific and general-domain. This demonstrates the effectiveness of the proposed reasoning strategies.
>
> | Dataset           |  CoNLL03     | WNUT17       | BC5CDR       | BioNLP11     | CRAFT        |
> |-------------------|--------------|--------------|--------------|--------------|--------------|
> | Vanilla           | **69.42** (0.91) | **46.61** (2.97) | 61.28 (3.11) | 51.29 (2.48) | 21.66 (1.41) |
> | Decomposed-QA     | 59.67 (0.36) | 42.39 (1.99) | **65.45** (0.89) | **55.3** (0.54)  | **23.99** (2.65) |
> | Syn.              |              |              |              |              |              |
> | Front             |              |              |              |              |              |
> | Noun phrases      | **57.13** (0.41) | **39.75** (2.42) | 64.41 (1.74) | 52.73 (1.89) | 22.92 (3.04) |
> | POS tag           | 55.14 (0.98) | 39.74 (1.98) | **66.24** (2.40) | 53.97 (0.99) | **23.59** (2.01) |
> | Constituency tree | 56.36 (1.07) | 39.66 (1.51) | 64.7 (0.80)  | 53.98 (1.24) | 23.84 (1.94) |
> | Dependency tree   | 54.27 (1.54) | 38.36 (0.37) | 65.56 (1.20) | **54.37** (0.83) | 23.51 (2.06) |
> | Back              |              |              |              |              |              |
> | Noun phrases      | **58.89** (1.39) | 36.47 (1.10) | **60.44** (0.57) | 51.99 (1.27) | 21.71 (1.89) |
> | POS tag           | 56.12 (1.54) | **38.12** (0.55) | 58.19 (1.55) | 54.41 (2.62) | 22.69 (3.76) |
> | Constituency tree | 55.66 (2.17) | 37.75 (2.04) | 47.81 (1.99) | 43.58 (1.44) | 23.86 (1.30) |
> | Dependency tree   | 58.36 (1.11) | 37.49 (1.56) | 59.69 (0.76) | **55.23** (0.77) | **22.84** (1.59) |
> | Tool.             |              |              |              |              |              |
> | POS tag           | **62.79** (2.54) | 43.81 (2.15) | **66.4** (1.44)  | **52.38** (0.35) | 24.54 (3.87) |
> | Constituency tree | 60.96 (0.34) | **44.3** (1.02)  | 65.02 (3.00) | 51.44 (0.81) | 23.88 (2.24) |
> | Dependency tree   | 59.23 (3.13) | 41.6 (1.30)  | 62.79 (2.62) | 42.71 (0.73) | **24.86** (2.45) |
> | Tool. + Syn.      |              |              |              |              |              |
> | Front             |              |              |              |              |              |
> | POS tag           | **63.46** (1.02) | 43.9 (2.21)  | 64.24 (1.83) | 49.87 (0.85) | **25.05** (2.12) |
> | Constituency tree | 59.59 (0.99) | **44.68** (2.59) | **65.43** (2.51) | **50.6** (1.39)  | 24.5 (2.76)  |
> | Dependency tree   | 57.93 (2.05) | 40.96 (3.57) | 59.93 (1.88) | 40.18 (1.63) | 24.46 (2.50) |
> | Back              |              |              |              |              |              |
> | POS tag           | 58.08 (0.27) | **39.32** (4.42) | **60.84** (3.47) | **47.9** (1.88)  | 13.71 (1.90) |
> | Constituency tree | **58.95** (2.19) | 37.91 (5.46) | 54.57 (1.86) | 42.27 (2.78) | 20.64 (1.85) |
> | Dependency tree   | 54.68 (1.27) | 36.7 (3.07)  | 57.28 (1.37) | 45.62 (3.46) | **24.05** (4.05) |
>
> [1] Wei C H, Peng Y, Leaman R, et al. Assessing the state of the art in biomedical relation extraction: overview of the BioCreative V chemical-disease relation (CDR) task[J]. Database, 2016, 2016: baw032.
>
> [2] Pyysalo S, Ohta T, Rak R, et al. Overview of the ID, EPI and REL tasks of BioNLP Shared Task 2011[C]//BMC bioinformatics. BioMed Central, 2012, 13: 1-26.
>
> [3] Bada M, Eckert M, Evans D, et al. Concept annotation in the CRAFT corpus[J]. BMC bioinformatics, 2012, 13(1): 1-20.
>
> [4] Crichton G, Pyysalo S, Chiu B, et al. A neural network multi-task learning approach to biomedical named entity recognition[J]. BMC bioinformatics, 2017, 18(1): 1-14.
>
> **Question C: Other LLMs.**
>
> To achieve similar complex reasoning and instruction comprehension abilities, the model's size and training techniques are generally required to be similar to ChatGPT. However, we currently cannot afford to run such a large model locally. So, we focus on the evaluation of ChatGPT. As a powerful but closed-source model, the assessment of ChatGPT is important for this community. And we leave the evaluation of other LLMs to our future work.
>
> **Question D: Potential ways to reduce the gap between fully-supervised SOTA.**
>
> We believe that the gap can be further reduced from the following aspects:
>
> (a) Combining different syntactic information to achieve enhanced reasoning abilities
>
> (b) Adding entity type information.
>
> (c) Exploring label logic and combinations in decomposed-QA.
>
> (d) Using parsing tools with better support for different languages, or combining information from different parsing tools.
>
> (e) Utilizing a small number of labeled demonstrations and exploring effective few-shot learning strategies to improve performance.
>
> For (a)-(d), please refer to responses to Question E for preliminary results.
>
> For (e), please refer to responses to Question A for preliminary results.
>
> **Question E: Next steps to address limitations and refine reasoning techniques.**
>
> We believe that reasoning techniques can be improved from the following aspects:
>
> (a) Combining different syntactic information to achieve enhanced reasoning abilities. This might alleviate the negative impact of the noisy syntactic information. The following table shows the preliminary experimental results of combining four types of syntactic information. The results show that the ensemble effectively improve the performance.
>
> |                     | Ontonotes 4  | PowerPlantFlat |
> |----------------------------|--------------|----------------|
> | Vanilla                    | 35.16 (1.57) | 27.85          |
> | Tool. + Syn. + Consistency |              |                |
> |   Word segmentation        | 42.46 (2.20) | 40.31          |
> |   POS tag                  | 40.86 (2.48) | 38.21          |
> |   Constituency tree        | 41.36 (3.58) | 35.76          |
> |   Dependency tree          | 40.49 (3.49) | 39.97          |
> | Ensemble                   | **43.26** (1.99) | **44.01**          |
>
> (b) Adding entity type information. Providing an introduction to entity types to enhance the model's understanding of each type. The following table shows the preliminary experimental results. The results show that the effectiveness of this strategy varies on different datasets. We will explore this in detail in our future work.
>
> |        | Ontonotes 4  | PowerPlantFlat |
> |---------------|--------------|----------------|
> | Vanilla       | 35.16 (1.57) | **27.85**          |
> |   + type info | **43.81** (3.89) | 25.78          |
> | Decomposed-QA | 38.79 (1.66) | **36.57**          |
> |   + type info | **44.39** (2.12) | 35.87          |
>
> (c) Exploring label logic and combinations in decomposed-QA. There might exist different relations between labels. Combining similar/related labels in decomposed-QA might achieve further improvements. Here, we conduct preliminary experiments. Below are the used one-by-one label orders and label group orders of Ontonotes 4 and PowerPlantFlat. We show them all in English for readability.
> * Ontonotes 4:
>   * One-by-one order: \[[Person], [Location], [Organization], [Geo-political entity]]
>   * Label group order: \[[Person, Organization], [Geo-political entity, Location]]
> * PowerPlantFlat:
>   * One-by-one order: \[[Device Identity], [Device Name], [System Identity], [System Name], [Component Name], [Location], [Person]]
>   * Label group order: \[[Device Identity, Device Name], [System Identity, System Name], [Component Name], [Location], [Person]]
>
> The results below reveal that appropriate label combinations can also achieve superior performance in decomposed-QA paradigm.
>
> | Dataset           | Ontonotes 4  | PowerPlantFlat |
> |-------------------|--------------|----------------|
> | Vanilla           | 35.16 (1.57) | 27.85          |
> | QA - one-by-one   | 38.79 (1.66) | 36.57          |
> | QA - label combinations | 40.01 (1.58) | 36.37          |
>
> (d) Using parsing tools with better support for different languages, or combining information from different parsing tools. Here, we evaluate another parsing tool, stanza, from Stanford University. To exclude the influence of other reasoning techniques and only observe the initial effects of different parsing tools, we conduct tool augmentation on four types of syntactic information under the vanilla paradigm. The results are shown in the table below.
>
> | Dataset (Language) | ACE05 (English) |              | PowerPlantFlat (Chinese) |        |
> |--------------------|-----------------|--------------|--------------------------|--------|
> | Parseing tool      | Hanlp           | Stanza       | Hanlp                    | Stanza |
> | Word segmentation  | \               | \            | **31.76**                    | 28.55  |
> | POS tag            | 30.71 (1.43)    | **31.51** (0.97) | **25.91**                    | 20.6   |
> | Constituency tree  | 28.06 (2.03)    | **28.64** (1.08) | **21.39**                    | 19.76  |
> | Dependency tree    | 28.32 (1.70)    | **28.38** (1.76) | **23.52**                   | 20.97  |
>
> The results show that the quality of parsing tools and their help in reasoning are related to the language used. Stanza, from English developers, has higher support for the English dataset, ACE05; Hanlp, from Chinese developers, has better support for the Chinese dataset, PowerPlantFlat. Therefore, when using our methods, we can choose parsing tools that are more suitable for the corresponding language.
>
> **Reasons To Reject: The augmentation strategies proposed lack significant novelty.**
>
> To the best of our knowledge, our work is the first to comprehensively adapt these prevalent reasoning techniques to zero-shot NER. Also, our work is the first to use syntactic augmentation approaches in the NER task with large language models.
>
> **Reasons To Reject: Not including standard chain-of-thought (CoT) methods.**
>
> Our work aims to present an empirical study of ChatGPT's reasoning capabilities on zero-shot NER. However, standard CoT methods belong to few-shot methods, which are beyond the scope of this work. Besides, we claimed in our paper that our proposed methods achieve remarkable improvements on zero-shot NER, which is supported by the results in Table 1&2.
>
> **Reasons To Reject: Not evaluating on datasets CoNLL-2003 and WNUT-17.**
>
> We have evaluated our proposed strategies on seven datasets in our paper, as shown in Tables 1&2. These datasets include both Chinese and English datasets and both domain-specific and general-domain scenarios. The experimental results fully support our conclusion that our proposed reasoning techniques achieve significant improvements on zero-shot NER.
>
>
> **Missing reference**
>
> This work has been cited in Line 132 of Section 2 Related Work in our paper.

---

### Meta-Review · Area_Chair_Evkk · 2023-09-20

**Recommendation:** 4

**Metareview:**

This paper proposes a clever adaptation of recent advances in LLMs - task decomposition and step-by-step reasoning to the task of NER. Various strategies are used to incorporate syntax prediction as an intermediate step and break down the eventual decision making problem into stage-wise decisions and their impact is carefully studied. All the reviewers like this paper. There was an extensive discussion on this paper and more analyses were provided by the authors in support of their results. These new results look convincing to me.

---

### Decision · Program_Chairs · 2023-10-07

**Decision:**

Accept-Main

**Comment:**

This paper proposes a clever adaptation of recent advances in LLMs - task decomposition and step-by-step reasoning to the task of NER. Various strategies are used to incorporate syntax prediction as an intermediate step and break down the eventual decision making problem into stage-wise decisions and their impact is carefully studied. All the reviewers like this paper. There was an extensive discussion on this paper and more analyses were provided by the authors in support of their results. These new results look convincing to me.